# Sophia: A Scalable Stochastic Second-order Optimizer for Language Model Pre-training

**Hong Liu    Zhiyuan Li    David Hall    Percy Liang    Tengyu Ma**

Stanford University
`{hliu99, zhiyuanli, dlwh, pliang, tengyuma}@cs.stanford.edu`

## Abstract

Given the massive cost of language model pre-training, a non-trivial improvement of the optimization algorithm would lead to a material reduction on the time and cost of training. Adam and its variants have been state-of-the-art for years, and more sophisticated second-order (Hessian-based) optimizers often incur too much per-step overhead. In this paper, we propose Sophia, **S**econd-**o**rder Cli**p**ped Stoc**h**astic Opt**i**miz**a**tion, a simple scalable second-order optimizer that uses a light-weight estimate of the diagonal Hessian as the pre-conditioner. The update is the moving average of the gradients divided by the moving average of the estimated Hessian, followed by element-wise clipping. The clipping controls the worst-case update size and tames the negative impact of non-convexity and rapid change of Hessian along the trajectory. Sophia only estimates the diagonal Hessian every handful of iterations, which has negligible average per-step time and memory overhead. On language modeling with GPT models of sizes ranging from 125M to 1.5B, Sophia achieves a 2x speed-up compared to Adam in the number of steps, total compute, and wall-clock time, achieving the same perplexity with 50% fewer steps, less total compute, and reduced wall-clock time.

## 1 Introduction

Language models (LLMs) have gained phenomenal capabilities as their scale grows (Radford et al., 2019; Kaplan et al., 2020; Brown et al., 2020; Zhang et al., 2022b; Touvron et al., 2023; OpenAI, 2023). However, pre-training LLMs is incredibly time-consuming due to the massive datasets and model sizes—hundreds of thousands of updates to parameters are required. For example, PaLM was trained for two months on 6144 TPUs, which costed 10 million dollars (Chowdhery et al., 2022).

Pre-training efficiency is thus a major bottleneck in scaling up LLMs. This work aims to improve pre-training efficiency with a faster optimizer, which either reduces the time and cost to achieve the same pre-training loss, or alternatively achieves better pre-training loss with the same budget.

Adam (Kingma & Ba, 2014) (or its variants (Loshchilov & Hutter, 2017; Shazeer & Stern, 2018; You et al., 2019)) is the dominantly used optimizer for training LLMs, such as GPT (Radford et al., 2019; Brown et al., 2020), OPT (Zhang et al., 2022b), Gopher (Rae et al., 2021) and LLAMA (Touvron et al., 2023). Designing faster optimizers for LLMs is challenging. First, the benefit of the first-order (gradient-based) pre-conditioner in Adam is not yet well understood (Liu et al., 2020; Zhang et al., 2020; Kunstner et al., 2023). Second, the choice of pre-conditioners is constrained because we can only afford light-weight options whose overhead can be offset by the speed-up in the number of iterations. On the other hand, Chen et al. (2023) automatically search among the light-weight gradient-based pre-conditioners and identify Lion, which is substantially faster than Adam on vision Transformers and diffusion models but only achieves limited speed-up on LLMs (Chen et al., 2023).

This paper introduces Sophia, **S**econd-**o**rder Cli**p**ped Stoc**h**astic Opt**i**miz**a**tion, a light-weight second-order optimizer that uses an inexpensive stochastic estimate of the diagonal of the Hessian as a pre-conditioner and a clipping mechanism to control the worst-case update size. On pre-training language models such as GPT-2, Sophia achieves the same validation pre-training loss with 50% fewer number of steps than Adam. Because Sophia maintains almost the memory and average time per step, the speedup also translates to 50% less total compute and 50% less wall-clock time (See Figure 1 (a)&(b)). We also note that comparing the run-time to achieve the same loss is the correct way to compare the speed of optimizers for LLMs; see Section 3.2 for more details.

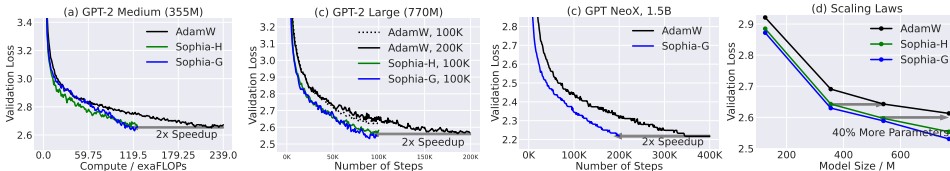

Figure 1: Sophia achieves significant speedup over AdamW in GPT-2 pre-trained on OpenWebText and GPT NeoX pre-trained on the Pile. (a) (b) (c) Comparison of the number of steps needed to achieve the same validation loss on (a) GPT-2-medium (355M), (b) GPT-2-large (770M) and (c) GPT NeoX 1.5B. Across all model sizes, Sophia needs 50% less time to reach the same validation loss as AdamW. (d) Validation losses of models with different sizes pre-trained. The gap between Sophia and AdamW gets larger as models size grows. Notably, a 540M-parameter model pre-trained with Sophia for 100K steps has the same loss as a 770M-parameter model pre-trained with AdamW for 100K steps. See Section 3 for details and more results.

---

**Algorithm 1** Hutchinson($\theta$)

1: **Input:** parameter $\theta$.
2: Compute mini-batch loss $L(\theta)$.
3: Draw $u$ from $\mathcal{N}(0, \mathrm{I}_d)$.
4: **return** $u \odot \nabla(\langle \nabla L(\theta), u \rangle)$.

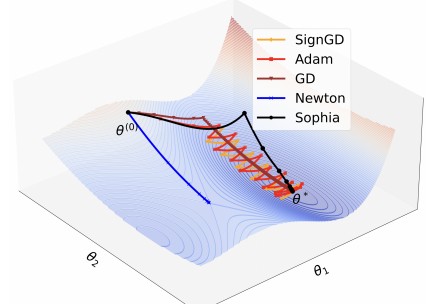

Figure 2: The motivating toy example. $\theta_{[1]}$ is the sharp dimension and $\theta_{[2]}$ is the flat dimension. GD's learning rate is limited by the sharpness in $\theta_1$. It makes slow progress along $\theta_{[2]}$. Adam and SignGD bounce along $\theta_{[1]}$ while making slow progress along $\theta_{[2]}$. Newton's method converges to a saddle point. Sophia makes fast progress in $\theta_{[1]}$ and $\theta_{[2]}$ and converges to the minimum with a few steps.

**Algorithm 2** Gauss-Newton-Bartlett (GNB)($\theta$)

1: **Input:** parameter $\theta$.
2: Draw a mini-batch of input $\{x_b\}_{b=1}^B$.
3: Compute logits on a batch: $\{f(\theta, x_b)\}_{b=1}^B$.
4: Sample $\hat{y}_b \sim \text{softmax}(f(\theta, x_b)), \forall b \in [B]$.
5: Calculate $\hat{g} = \nabla(1/B \sum \ell(f(\theta, x_b), \hat{y}_b))$.
6: **return** $B \cdot \hat{g} \odot \hat{g}$.

---

**Algorithm 3** Sophia

1: **Input:** $\theta_1$, learning rate $\{\eta_t\}_{t=1}^T$, hyperparameters $\lambda, \beta_1, \beta_2, \epsilon$, and estimator choice Estimator $\in \{$Hutchinson,GNB$\}$
2: Set $m_0 = 0$, $v_0 = 0$, $h_{1-k} = 0$
3: **for** $t = 1$ **to** $T$ **do**
4:     Compute minibach loss $L_t(\theta_t)$.
5:     Compute $g_t = \nabla L_t(\theta_t)$.
6:     $m_t = \beta_1 m_{t-1} + (1 - \beta_1) g_t$
7:     **if** $t \bmod k = 1$ **then**
8:         Compute $\hat{h}_t = $ Estimator$(\theta_t)$.
9:         $h_t = \beta_2 h_{t-k} + (1 - \beta_2) \hat{h}_t$
10:    **else**
11:        $h_t = h_{t-1}$
12:    $\theta_t = \theta_t - \eta_t \lambda \theta_t$ (weight decay)
13:    $\theta_{t+1} = \theta_t - \eta_t \cdot \text{clip}(m_t / \max\{\gamma \cdot h_t, \epsilon\}, 1)$

---

Moreover, the scaling law based on model size from 125M to 770M is in favor of Sophia over Adam—the gap between Sophia and Adam with 100K steps increases as the model size increases (Figure 1 (c)). In particular, Sophia on a 540M-parameter model with 100K steps gives the same validation loss as Adam on a 770M-parameter model with 100K steps.

Sophia estimates the diagonal entries of the Hessian of the loss using a mini-batch of examples every $k$ step (with $k = 10$ in practice). We consider two options for diagonal Hessian estimators: (a) an unbiased estimator that uses a Hessian-vector product, and (b) a biased estimator that uses one mini-batch gradient calculated with resampled labels. Both the two estimators only introduce 5% overheads per step (in average). At every step, Sophia updates the parameter with an exponential moving average (EMA) of the gradient divided by the EMA of the diagonal Hessian estimate, subsequently clipped by a scalar. (All operations are element-wise.) See Algorithm 3 for the pseudo-code.

Sophia has a more aggressive pre-conditioner—it applies a stronger penalization to updates in sharp dimensions (where the Hessian is large) than the flat dimensions (where the Hessian is small), ensuring a uniform *loss* decrease across all parameter dimensions. In contrast, Adam's updates are mostly uniform across all parameter dimensions, leading to a slower loss decrease in flat dimensions. (See Section 2.1 for more discussions.) These make Sophia converge in fewer iterations. Thanks to the light-weight diagonal Hessian estimate, the speed-up in the number of steps translates to a speed-up in total compute or wall-clock time.

Sophia's clipping mechanism controls the worst-case size of the updates in all directions, safeguarding against the negative impact of inaccurate Hessian estimates, rapid Hessian changes, and non-

convex landscape (with which Newton's method may converge to local maxima or saddle points instead of local minima). The safeguard allows us to estimate Hessian infrequently (every $k = 10$ steps). In contrast, prior second-order methods often update Hessian estimates every step.

## 2 METHOD

We instantiate gradient descent and Adam on a simplified 2D problem to motivate the use of second-order information and per-coordinate clipping in Section 2.1. We present Sophia in Section 2.2, with pseudo-code in Algorithm 3. We introduce two estimators of diagonal Hessian in Section 2.3.

### 2.1 MOTIVATIONS

**Heterogeneous curvatures.** The loss functions in deep learning often have different curvatures across different parameter dimensions (Sagun et al., 2016; Ghorbani et al., 2019; Zhang et al., 2020; Yao et al., 2020). We demonstrate the limitations of Adam and GD on heterogeneous landscapes in a two-dimensional loss function $L(\theta_{[1]}, \theta_{[2]}) = L_1(\theta_{[1]}) + L_2(\theta_{[2]})$ where $L_1$ is much sharper than $L_2$. We plot the loss landscape of $L(\theta_{[1]}, \theta_{[2]})$ in Figure 2.[1] For simplicity, we discuss GD and deterministic versions of Adam. GD's update in this setting is:

$$\theta_{[1]} \leftarrow \theta_{[1]} - \eta \cdot L_1'(\theta_{[1]}) \text{ and } \theta_{[2]} \leftarrow \theta_{[2]} - \eta \cdot L_2'(\theta_{[2]}). \tag{1}$$

A common simplification of Adam that is more amenable to analysis (Balles & Hennig, 2018; Bernstein et al., 2018; Zhuang et al., 2020; Kunstner et al., 2023) is SignGD, which dates back to RProp (Braun & Riedmiller, 1992) that motivated RMSProp (Hinton et al., 2012) and Adam. Without using the EMA (for both the gradient and second moments of the gradient), Adam's update is simplified to $\eta \cdot \nabla L(\theta)/|\nabla L(\theta)| = \eta \cdot \text{sign}(\nabla L(\theta))$, which is called SignGD:

$$\theta_{[1]} \leftarrow \theta_{[1]} - \eta \cdot \text{sign}(L_1'(\theta_{[1]})) \text{ and } \theta_{[2]} \leftarrow \theta_{[2]} - \eta \cdot \text{sign}(L_2'(\theta_{[2]})). \tag{2}$$

**Limitations of GD and SignGD (Adam).** The optimal learning rate of GD should be proportional to the inverse of the curvature. Let $h_1$ and $h_2$ be the curvatures of $L_1$ and $L_2$ at the local minimum (and thus $h_1 \gg h_2$). The optimal learning rate for $\theta_{[1]}$ in (1) is $\asymp 1/h_1$, which is much smaller than the optimal learning rate that $\theta_{[2]}$ needs ($\asymp 1/h_2$). The largest shared learning rate can only be $1/h_1$; consequently, the convergence in $\theta_{[2]}$ is slow as demonstrated in the brown curve of Figure 2.

The update size of SignGD equals learning rate $\eta$ in all dimensions. The same update size translates to less progress in the flat direction than in the sharp direction. As shown in the yellow curve of Figure 2, the progress of SignGD in $\theta_{[2]}$ is slow because each step only decreases the loss $L_2(\theta_{[2]})$ slightly. On the other hand, along $\theta_{[1]}$, the iterate quickly goes to the valley in the first three steps and then starts to bounce. To fully converge in the sharp $\theta_{[1]}$, the learning rate $\eta$ needs to decay to 0, which will exacerbate the slow convergence in the flat $\theta_{[2]}$. The trajectory of Adam in this example is similar to SignGD, which is plotted as the red curve in Figure 2.

The behavior of SignGD and Adam calls for more aggressive pre-conditioning—sharp dimensions should have relatively smaller updates than flat dimensions so that the decrease of loss is equalized in all dimensions. As suggested by literature on second-order optimization (Boyd & Vandenberghe, 2004) for convex functions, the optimal pre-conditioner should be the Hessian, which captures the curvature of each dimension; the update is the gradient divided by the Hessian of each dimension:

$$\theta_{[1]} \leftarrow \theta_{[1]} - \eta \cdot L_1'(\theta_{[1]})/h_1 \text{ and } \theta_{[2]} \leftarrow \theta_{[2]} - \eta \cdot L_2'(\theta_{[2]})/h_2. \tag{3}$$

**Limitations of Newton's method.** For non-convex functions, Newton's method may not converge to a minimum. In the blue curve of Figure 2, Newton's method quickly converges to a saddle point. The curvature might also change rapidly along the trajectory, making the second-order information unreliable. To address these limitations, we propose considering only pre-conditioners that capture positive curvature, and introduce pre-coordinate clipping to mitigate the rapid change of Hessian (more detail in Section 2.2). Applying our algorithm on the toy case results in the following update:

$$\theta_{[1]} \leftarrow \theta_{[1]} - \eta \cdot \text{clip}(L_1'(\theta_{[1]})/\max\{h_1, \epsilon\}, \rho) \text{ and } \theta_{[2]} \leftarrow \theta_{[2]} - \eta \cdot \text{clip}(L_2'(\theta_{[2]})/\max\{h_2, \epsilon\}, \rho), \tag{4}$$

where $\rho$ is a constant to control the update size, $\epsilon$ is a very small constant (e.g., 1e-12), which avoids dividing by 0. When the curvature of some dimension is changing rapidly or negative and thus the second-order information is misleading, the clipping mechanism kicks in and the optimizer defaults

---

[1]Concretely, in Figure 2, $L_1(\theta_{[1]}) = 8(\theta_{[1]} - 1)^2(1.3\theta_{[1]}^2 + 2\theta_{[1]} + 1)$ and $L_2(\theta_{[2]}) = 1/2(\theta_{[2]} - 4)^2$.

to SignGD. Numerous prior methods such as trust region (Conn et al., 2000), backtracking line search (Boyd & Vandenberghe, 2004), and cubic regularization (Nesterov & Polyak, 2006) tackle the same issue, but the clipping mechanism is simpler and more efficient.

As shown in the black curve in Fig. 2, the update in equation (4) starts off similarly to SignGD due to the clipping mechanism in the non-convex region, making descent opposed to converging to the saddle point. Then, in the convex valley, it converges to the global minimum with a few steps. Compared with SignGD and Adam, it makes much faster progress in the flat dimension $\theta_{[2]}$, while avoiding boucing in the sharp dimension $\theta_{[1]}$.

## 2.2 Sophia: Second-order Clipped Stochastic Optimization

Section 2.1 demonstrates Adam does not sufficiently adapt to the heterogeneous curvatures. On the other hand, Newton's method has a pre-conditioner optimal for convex functions, but is vulnerable to negative curvature and rapid change of Hessian. With these insights, we design a new optimizer, Sophia, which is more adaptive to heterogeneous curvatures than Adam, more resistant to non-convexity and rapid change of Hessian than Newton's method, and uses a low-cost pre-conditioner.

We use $\theta_t$ to denote the parameter at time step $t$. At each step, we sample a mini-batch from the data distribution and calculate the mini-batch loss, denoted by $L_t(\theta_t)$. We denote by $g_t$ the gradient of $L_t(\theta_t)$, $g_t = \nabla L_t(\theta_t)$. Let $m_t$ be the EMA of gradients, $m_t \leftarrow \beta_1 m_{t-1} + (1 - \beta_1)g_t$.

**EMA of diagonal Hessian estimates.** Sophia uses a diagonal Hessian pre-conditioner, which directly adjusts the update size of different parameters according to their curvatures. We will present two efficient estimators of diagonal Hessian in Section 2.3. To mitigate the overhead, we only estimate the Hessian every $k$ steps ($k = 10$ in practice). At time step $t$ with $t \bmod k = 1$, the estimator returns an estimate $\hat{h}_t$ of the diagonal of the Hessian of the mini-batch loss.

Similar to the gradient of the mini-batch loss function, the estimated diagonal Hessian can also have large noise. Inspired by the EMA of moments of gradients in Adam, we also denoise the diagonal Hessian estimates with EMA across iterations. We update the EMA every $k$ steps, resulting in the following update rule for the diagonal Hessian estimate:

$$h_t = \beta_2 h_{t-k} + (1 - \beta_2)\hat{h}_t \text{ if } t \bmod k = 1; \text{ else } h_t = h_{t-1}. \tag{5}$$

**Per-coordinate clipping.** As discussed in Section 2.1, on nonconvex functions, vanilla Newton's method, which uses Hessian as the pre-conditioner, may converge to local maxima instead of local minima. In addition, the inaccuracy of Hessian estimates and the change of Hessian along the trajectory can make the second-order information unreliable. To this end, we (1) only consider the positive entries of the diagonal Hessian and (2) introduce per-coordinate clipping to the update. For a clipping threshold $\rho > 0$, let the clipping function be $\text{clip}(z, \rho) = \max\{\min\{z, \rho\}, -\rho\}$ where all operations are applied coordinate-wise. The update rule is written as:

$$\theta_{t+1} \leftarrow \theta_t - \eta_t \cdot \text{clip}(m_t / \max\{\gamma \cdot h_t, \epsilon\}, 1), \tag{6}$$

where $\epsilon > 0$ is a small constant to avoid dividing by 0, and $\gamma$ controls the fraction of clipped entries. We present the pseudo-code of the Sophia in Algorithm 3.

If $h_t[i] < 0$, the corresponding entry in the pre-conditioned gradient $m_t[i]/\max\{\gamma \cdot h_t[i], \epsilon\} = m_t[i]/\epsilon$ is large and has the same sign as $m_t[i]$, and thus $\eta \cdot \text{clip}(m_t[i]/\max\{\gamma \cdot h_t[i], \epsilon\}, 1) = \eta \cdot \text{sign}(m_t[i])$, which is the same as stochastic momentum SignSGD. In other words, Sophia uses stochastic Sign Momentum GD as a backup when the Hessian is negative (or mistakenly estimated to be negative.) Also note that clipping controls the worst-case update size in all parameter dimensions to be at most $\rho$, which also improves stability. Moreover, because for many parameter dimensions, the clipping is not activated and the update is automatically adjusted, our worst-case update size $\eta$ can be chosen to be larger than the worst update size $\eta$ in stochastic Sign Momentum GD.

Several previous works (Becker & Le Cun, 1988; Chapelle et al., 2011; Schaul et al., 2013; Yao et al., 2021) use diagonal Hessian as a pre-conditioner in optimizers for training neural networks. However, they use more frequent Hessian estimations, which leads to significant per-step computation overhead, most likely because of the lack of the clipping mechanism that safeguards against inaccurate and changing Hessian. In general, to the best of our knowledge, there has not been previous reports that showed second-order optimizers achieve a speed-up on decoder-only large language models in wall-clock time or total compute (see more related work and discussions in Section A).

## 2.3 DIAGONAL HESSIAN ESTIMATORS

We introduce two diagonal Hessian estimators, both of which have memory and run-time costs similar to computing a gradient (up to constant factors).

**Option 1: Hutchinson's unbiased estimator.** For any loss function $\ell(\theta)$ on parameters $\theta \in \mathbb{R}^d$, the Hutchinson's estimator (Hutchinson, 1989; Roosta-Khorasani & Ascher, 2015; Yao et al., 2021) first draws $u \in \mathbb{R}^d$ from the spherical Gaussian distribution $\mathcal{N}(0, I_d)$, and then outputs $\hat{h} = u \odot (\nabla^2 \ell(\theta) u)$, where $\odot$ denotes the element-wise product, and $\nabla^2 \ell(\theta) u$ is the product of the Hessian with the vector $u$. The Hutchinson's estimator is an unbiased estimator for the diagonal of the Hessian, because $\mathbb{E}[\hat{h}] = \mathrm{diag}(\nabla^2 \ell(\theta))$. It only requires a Hessian-vector product (i.e., $\nabla^2 \ell(\theta) u$), which have efficient implementations in PyTorch and JAX, instead of the full Hessian matrix.

**Option 2: Gauss-Newton-Bartlett (GNB) estimator.** Suppose $\ell(\theta, (x, y))$ is a loss function on an example $(x, y)$ of the form $\ell(\theta, (x, y)) = \ell_{ce}(f(\theta, x), y)$ where $\ell_{ce}$ is cross-entropy and $f(\theta, x) \in \mathbb{R}^V$ is the logits, and $V$ is the number of items/classes in a multi-class classification problem (e.g., the vocabulary size in LLMs). First, the Hessian of $\ell(\theta, (x, y))$ (w.r.t to variable $\theta$) has the well-known Gauss-Newton (GN) decomposition (Ortega & Rheinboldt, 2000; Schraudolph, 2002),

$$\nabla^2_\theta \ell(\theta) = J_\theta f(\theta, x) \cdot S \cdot J_\theta f(\theta, x)^\top + J_{\theta\theta} f(\theta, x)[q] \tag{7}$$

where $J_\theta f(\theta, x)$ is the Jacobian of $f$ w.r.t to $\theta$ viewed as a matrix in $\mathbb{R}^{d \times V}$, $S = \partial^2 \ell_{ce}(t, y)/\partial t^2 \big|_{t=f(\theta,x)} \in \mathbb{R}^{V \times V}$ is the second-order derivatives of the loss w.r.t the logits, $q = \partial \ell_{ce}(t, y)/\partial t \big|_{t=f(\theta,x)} \in \mathbb{R}^V$ is the first-order derivatives of the loss w.r.t the logits, and $J_{\theta\theta} f(\theta, x)$ is the second-order derivatives of the multi-variate function $f(\theta, x)$ w.r.t $\theta$, viewed as a linear map from $\mathbb{R}^V$ to $\mathbb{R}^{d \times d}$. In neural networks, past works found that the second term $J_{\theta\theta} f(\theta, x)[q]$ in (11) is often relative smaller than the first term $J_\theta f(\theta, x) \cdot S \cdot J_\theta f(\theta, x)^\top$ (Sankar et al., 2021), which is often referred to as the Gauss-Newton (GN) matrix (Dennis Jr & Schnabel, 1996; Ortega & Rheinboldt, 2000; Schraudolph, 2002; Chen, 2011) and used as pre-conditioners in second-order optimizers (Botev et al., 2017; Martens, 2020; Gargiani et al., 2020). Following these works, we build an unbiased estimator for *the diagonal* of the GN matrix, as a biased estimator of the Hessian diagonal.

First note that $S$ only depends $f(\theta, x)$ but not $y$, even though the loss depends on $y$.[2] Thus, $S = \partial^2 \ell_{ce}(t, \hat{y})/\partial t^2 \big|_{t=f(\theta,x)}$ for any $\hat{y} \in \{1, \ldots, V\}$, which implies that $S = \mathbb{E}_{\hat{y} \sim p(\theta,x)} \big[ \partial^2 \ell_{ce}(t, \hat{y})/\partial t^2 \big|_{t=f(\theta,x)} \big]$. Because $\ell_{ce}(t, y)$ is the negative log-probability of the probabilistic model defined by the categorical distribution $\mathrm{Cat}(t)$ with parameter $t$, by Bartlett's second identity (Bartlett, 1953), we have that, $S = \mathbb{E}_{\hat{y} \sim \mathrm{Cat}(t)} \big[ \partial^2 \ell_{ce}(t, \hat{y})/\partial t^2 \big] = \mathbb{E}_{\hat{y} \sim \mathrm{Cat}(t)} \big[ \partial \ell_{ce}(t, \hat{y})/\partial t (\partial \ell_{ce}(t, \hat{y})/\partial t)^\top \big]$, where the first equality holds for $t = f(\theta, x)$ and the second equality holds for all $t$ by Bartlett's second identity. Therefore, the GN matrix satisfies

$$J_\theta f(\theta, x) \cdot S \cdot J_\theta f(\theta, x)^\top = \mathbb{E}_{\hat{y} \sim \mathrm{Cat}(t)} \left[ J_\theta f(\theta, x) \partial \ell_{ce}(t, \hat{y})/\partial t (\partial \ell_{ce}(t, \hat{y})/\partial t)^\top J_\theta f(\theta, x)^\top \right]$$

$$= \mathbb{E}_{\hat{y} \sim \mathrm{Cat}(t)} \left[ \nabla_\theta \ell_{ce}(f(\theta, x), \hat{y}) \nabla_\theta \ell_{ce}(f(\theta, x), \hat{y})^\top \right], \tag{8}$$

implying $\mathrm{diag}(J_\theta f(\theta, x) \cdot S \cdot J_\theta f(\theta, x)^\top) = \mathbb{E}_{\hat{y} \sim \mathrm{Cat}(t)}[\nabla_\theta \ell_{ce}(f(\theta, x), \hat{y}) \odot \nabla_\theta \ell_{ce}(f(\theta, x), \hat{y})]$. Hence, the quantity $\ell_{ce}(f(\theta, x), \hat{y}) \odot \nabla_\theta \ell_{ce}(f(\theta, x), \hat{y})$ is an unbiased estimator of the GN matrix for the Hessian of a one-example loss $\ell(f(\theta, x), y)$.

*Mini-batch version.* Given a mini-batch of inputs $\{(x_b, y_b)\}_{b=1}^B$. The most natural way to build an estimator for the diagonal of the GN matrix for the Hessian of the mini-batch loss is using

$$\frac{1}{B} \sum_{b=1}^B \nabla \ell_{ce}(f(\theta, x_b), \hat{y}_b) \odot \nabla_\theta \ell_{ce}(f(\theta, x_b), \hat{y}_b), \tag{9}$$

where $\hat{y}_b$'s are labels sampled from the model on inputs $x_b$'s respectively. However, as noted by Grosse (2022), implementing this estimator is inconvenient under the current auto-differentiation frameworks, where the users only have access to the average gradient over a mini-batch (as opposed

---

[2]Let $p(\theta, x) = \mathrm{softmax}(f(\theta, x)) \in \mathbb{R}^V$ the probability vector obtained from softmax of logits. $S = \mathrm{diagonal}(p(\theta, x)) - p(\theta, x)p(\theta, x)^\top$, where $\mathrm{diagonal}(z)$ is the matrix with vector $z$ residing on the diagonal.

to the individual ones). Fortunately, by Bartlett's first identity (Bartlett, 1953) (which generally holds for negative log-likelihood loss of probabilistic models), we have: $\forall b, \ \mathbb{E}_{\hat{y}_b} \nabla \ell_{ce}(f(\theta, x_b), \hat{y}_b) = 0$. Let $\widehat{L}(\theta) = \frac{1}{B} \sum_{b=1}^{B} \ell_{ce}(f(\theta, x_b), \hat{y}_b)$ be the mini-batch loss on the *sampled* labels (as opposed to the original labels). Observing that $\hat{y}_b$'s are independent with each other, we have

$$\mathbb{E}_{\hat{y}_b's} \left[ B \cdot \nabla_\theta \widehat{L}(\theta) \odot \nabla_\theta \widehat{L}(\theta) \right] = \mathbb{E}_{\hat{y}_b's} \left[ \frac{1}{B} \sum_{b=1}^{B} \nabla \ell_{ce}(f(\theta, x_b), \hat{y}_b) \odot \sum_{b=1}^{B} \nabla \ell_{ce}(f(\theta, x_b), \hat{y}_b) \right]$$

$$= \mathbb{E}_{\hat{y}_b's} \left[ \frac{1}{B} \sum_{b=1}^{B} \nabla \ell_{ce}(f(\theta, x_b), \hat{y}_b) \odot \nabla \ell_{ce}(f(\theta, x_b), \hat{y}_b) \right] \quad (10)$$

The RHS of (16) is the same as the expectation of (14), which by (13) also equals the diagonal of the GN matrix for the mini-batch loss. Hence, we use $B \cdot \nabla_\theta \widehat{L}(\theta) \odot \nabla_\theta \widehat{L}(\theta)$ as the estimator. To the best of our knowledge, Wei et al. (2020) first used this estimator. Given the use Bartlett's identities that are central to the estimator, we call it Gauss-Newton-Bartlett (GNB) estimator.

*GNB estimator for exponential family.* If $y$ is drawn from an exponential family $p(y; \eta)$ where the natural parameter $\eta$ is set to be $f(\theta, x)$ and the loss function $\ell(f(\theta, x), y)$ is the negative log-likelihood loss for the corresponding probabilistic distribution, then all the derivations still hold because (1) $S$ still only depends on $f(\theta, x)$ but not $y$, and (2) Bartlett's identities still hold.

*GNB estimator for squared loss.* When $y, f(\theta, x) \in \mathbb{R}$ and $\ell(f(\theta, x), y) = \frac{1}{2}(f(\theta, x) - y)^2$, the $S$ matrix is identity, and thus one can simply use $J_\theta f(\theta, x) J_\theta f(\theta, x)^\top$ as the estimator.

**Comparisons of Hessian estimators.** The Hutchinson's estimator does not assume any structure of the loss, but requires Hessian-vector products. The GNB estimator only estimates the GN term and always gives a positive semi-definite (non-negative) diagonal Hessian estimate. The PSDness ensures that the pre-conditioned update is always a descent direction (Dennis Jr & Schnabel, 1996).

## 3 EXPERIMENTS

We name the algorithm using Hutchinson's and GNB estimator Sophia-H and Sophia-G, respectively. We evaluate Sophia on GPT of sizes ranging from 125M to 6.6B. Results indicate that Sophia achieves the same or smaller validation loss than AdamW (Loshchilov & Hutter, 2017) in 50% less number of steps, total compute, and wall-clock time across different model sizes.

### 3.1 EXPERIMENTAL SETUP

**Language modeling.** We train autoregressive models on OpenWebText (Gokaslan & Cohen, 2019) and the Pile (Gao et al., 2020) from scratch. Following standard protocol, we set the context length of GPT-2 to 1024, and the context length of GPT-2 NeoX (Black et al., 2022) to 2048. We consider GPT-2 with 125M (small), 355M (medium), and 770M (large) parameters, and GPT NeoX with 1.5B and 6.6B parameters, respectively. Detailed model configurations are deferred to Section C.2.

**Baselines.** We compare Sophia with Adam with decoupled weight decay (AdamW) (Loshchilov & Hutter, 2017), the dominantly used optimizer on language modeling tasks, AdaHessian (Yao et al., 2021) which uses the EMA of the square of the diagonal Hessian estimate in its denominator, and Lion (Chen et al., 2023), a first-order adaptive optimizer discovered by symbolic search. For the 30M model, all hyperparameters are tuned with grid search. For other models, all hyperparmeters but the peak learning rate are configured as identical to those found on the 30M model. For models with size 125M and 355M, the peak learning rates are obtained through grid search. For larger models, we search for the largest possible peak learning rate such that the training does not blow up, and ensure 1.25 times the chosen learning rate will lead to a blow-up. For AdamW we found the well-established practice ($\beta_1 = 0.9$ and $\beta_2 = 0.95$) works consistently better than other choices. For Lion, we use $\beta_1 = 0.95$ and $\beta_2 = 0.98$ following Chen et al. (2023). For AdaHessian, we found $\beta_1 = 0.92$ and $\beta_2 = 0.99$ works the best. Details on hyperparameter tuning are deferred to Section C.1.

**Implementation.** We use batch size 480 for GPT-2 and 2048 for GPT NeoX. We use cosine LR schedule with the final LR equaling 0.05 times the peak LR with a fixed 2k steps of LR warm-up following Rae et al. (2021). We use standard gradient clipping (by norm) threshold 1.0. For Sophia, we use $\beta_1 = 0.96$, $\beta_2 = 0.99$, $\epsilon =$1e-12 and update diagonal Hessian every 10 steps. For Sophia-H, we use a subset of 32 examples from the minibatch to calculate the diagonal Hessian. For Sophia-G, we use a subset of 240 examples from the minibatch to calculate the diagonal Gauss-Newton. We

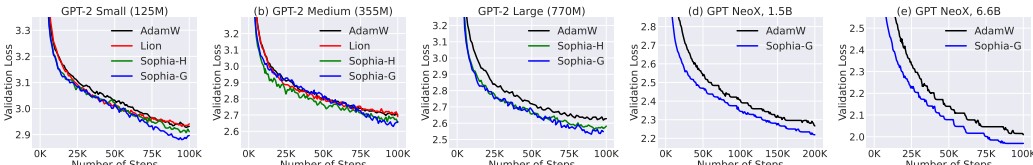

Figure 3: Comparison of numbers of steps to reach the same validation loss on OpenWebText. (a) Learning rate schedules. (b) GPT-2 Medium (355M). (c) GPT-2 Large (770M). (d) GPT NeoX 1.5B.

Figure 4: Validation loss. (a) GPT-2 Small (125M). AdamW: 2.921, Lion: 2.924, Sophia-H: 2.901, Sophia-G: 2.875 (b) GPT-2 Medium (355M). Adam: 2.691, Lion: 2.678, Sophia-H: 2.645, Sophia-G: 2.627. (c) GPT-2 Large (770M). AdamW: 2.613, Sophia-H: 2.554, Sophia-G: 2.524. (d) GPT NeoX 1.5B. AdamW: 2.250, Sophia-G: 2.218.(e) GPT NeoX 6.6B. AdamW: 1.992, Sophia-G: 1.969.

implement the algorithms in PyTorch (Paszke et al., 2019) and JAX (Bradbury et al., 2018) and train all the models in bfloat16. 125M and 355M models are trained on A5000 GPUs, while the 770M models are trained on A100 GPUs. We use a TPU v3-128 slice to train 1.5B and 6.6B GPT NeoX.

**Hyperparamter tuning strategy.** We refer to Section C.1 for the details on hyperparameters and only discuss two key hyperparameters, $\gamma$ and the peak learning rate $\eta$ in the main text. Similar to the protocol of baselines, all other hyperparameters are tuned on a 30M model and remain fixed for all the model sizes. For the peak learning rate and $\gamma$, we found the following strategy general works well, and delivers almost the same performance as those found by grid search.

- On a small model, tune $\gamma$ to make the proportion of coordinates where the update is not clipped (i.e., $|m_t / \max\{\gamma \cdot h_t, \epsilon\}| < 1$) in the range of $10\% - 50\%$. The same $\gamma$ likely can be transferred to models with the same architecture and data but different number of parameters. We use $\gamma = 0.01$ for Sophia-H and $\gamma = 0.05$ for Sophia-G in this paper.
- Suppose we already find a suitable $\gamma$ following the above procedure. We can then set the learning rate of Sophia to be either 3-5 times the learning rate that one would have used for Lion, or 0.8 times the learning rate that one would have used for AdamW.

## 3.2 EVALUATION

**Methodology for comparing the optimizers for LLMs.** One correct (and preferred) way of claiming optimizer $S$ is 2x faster than optimizer $A$ is comparing the following two experiments (for a variety of $T$'s). (1) running optimizer $A$ (e.g. Adam) with $T$ steps, with the optimal learning rate and learning rate schedule (tuned for running for $T$ steps) (2) running optimizer $S$ (e.g., Sophia) with $T/2$ steps, with any learning rate schedule, If Experiment 2 achieves a loss that is smaller than or equal to the loss of Experiment 1, then we say optimizer $S$ is 2x faster than optimizer $A$.

Note that modern learning rate schedulers such as cosine learning rate (Loshchilov & Hutter, 2016) are highly sensitive to a pre-specific total number of steps. Figure 3 (a) shows that with the same peak learning rate, the learning rate in a run with $T/2$ steps decays faster than that with $T$ steps. Moreover, the loss of the $T/2$-steps run decays faster initially than the $T$-steps run but ends up larger. The latter is also not a continuation of the former. Thus, in the proposed comparison above, we insist that Experiment 2 has a loss smaller than or equal to Experiment 1 without any approximations, because even "Adam with $T/2$ steps" can possibly achieve a loss similar to (but slightly worse than) "Adam with $T$ steps with the same peak learning rate". See Figure 3(b)-(d).

**Technical details.** Following the methodology above, we train baselines and Sophia for 100K, 200K, or 400K. We primarily evaluate the models with their log perplexity and plot the loss curves. We also report in-context learning results (with 2-shot exemplars and greedy decoding) on Super-GLUE (Wang et al., 2019). We average the results of 5 prompts (Section C.3).

## 3.3 RESULTS

Figure 4 illustrates the validation loss curve (log perplexity) on OpenWebText or the Pile with the same number of steps. Sophia-H and Sophia-G consistently achieve better validation loss than

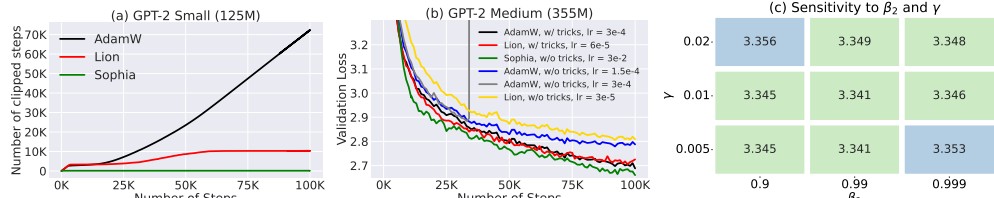

Figure 5: Few-shot evaluation on SuperGLUE. With the same number of steps, models pre-trained with Sophia outperforms models pre-trained with AdamW and Lion on most tasks. Models pre-trained with Sophia for 200K steps have comparable performance as models pre-trained with AdamW for 400K steps.

Figure 6: (a) AdamW and Lion trigger gradient clipping frequently. With Sophia, gradient clipping rarely happens. (b) AdamW and Lion require re-parameterizing attention with a temperature (inverse of layer index). The plot shows the largest LR that AdamW and Lion without the trick allow to be stable, which is much smaller than with the trick. In contrast, Sophia does not need this trick. (c) Sophia is not sensitive to $\gamma$ and $\beta_2$ choice.

AdamW, Lion, and AdaHessian. As the model size grows, the gap between Sophia and baselines also becomes larger. Sophia-H and Sophia-G both achieve a 0.05 smaller validation loss on the 355M / 770M model (Figure 4 (b) (c)) with the same 100K steps. This is a significant improvement since according to scaling laws in this regime (Kaplan et al., 2020; Hoffmann et al., 2022) and results in Figure 3, an improvement in loss of 0.05 is equivalent to 2x improvement in terms of number of steps or total compute to achieve the same validation loss.

**Sophia achieves the same loss in 50% less steps, total compute and wall-clock time.** The improvement in validation loss brought by Sophia can be translated into reduction of number of steps or total compute. In Figure 1 and Figure 3, we evaluate the optimizers by comparing the number of steps or total compute needed to achieve *the same or smaller validation loss*. Sophia achieve a 2x speedup compared with AdamW and Lion across different model sizes.

**The scaling law is in favor of Sophia over AdamW.** In Figure 1 (d), we plot the validation loss on OpenWebText of models of different sizes pre-trained for 100K steps. On OpenWebText, the gap between Sophia and Adam with 100K targeted steps is larger on 355M / 770M models than 125M models. Moreover, the 540M model trained by Sophia-H has smaller loss than the 770M model trained by AdamW. The 355M model trained by Sophia-H has comparable loss as the 540M model trained by AdamW.

**Few-shot Evaluation on SuperGLUE.** As shown in Figure 5, the improvement in validation loss transfers to downstream tasks. With the same number of steps in pre-training, GPT-2 and GPT NeoX pre-trained with Sophia have better few-shot accuracy on most subtasks. Models pre-trained with Sophia have comparable few-shot accuracy as models pre-trained with AdamW for 2x steps.

### 3.4 ANALYSIS

**Comparison of wall-clock time and amount of compute.** We compare the total compute (TFlops) per step and the wall-clock time on A100 GPUs in Table 1. We report average time per step (T(step)), time spent in Hessian computation (T(Hessian)) and total compute following Chowdhery et al. (2022). Since we calculate the diagonal Hessian estimate with a reduced batch size every 10 steps, the computation of the Hessian accounts for 6% of the total compute,

Table 1: Wall-clock time and compute.

| Algorithm | Model Size | T(step) | T(Hessian) | Compute |
|---|---|---|---|---|
| AdamW | 770M | 3.25s | – | 2550 |
| Sophia-H | 770M | 3.40s | 0.12s | 2708 |
| Sophia-G | 770M | 3.42s | 0.17s | 2678 |
| AdamW | 355M | 1.77s | – | 1195 |
| Sophia-H | 355M | 1.88s | 0.09s | 1249 |
| Sophia-G | 355M | 1.86s | 0.09s | 1255 |

and the overall wall-clock time overhead is less than 5% compared with AdamW. In terms of memory usage, our optimizer has two states, $m$ and $h$, which results in the same memory cost as AdamW.

**Sensitivity to $\rho$ and $\beta_2$, and transferability of hyperparameters.** On a 30M model, we perform a grid search to test the sensitivity of Sophia-H to hyperparamters (Figure 6 (c)). All combinations have a similar performance, while $\beta_2 = 0.99$ and $\gamma = 0.01$ performs the best. Moreover, this

Figure 7: (a) Hessian update frequency. (b) Diagonal Hessian pre-conditioners. (c) Element-wise clipping.

hyperparameter choice is transferable across model sizes. For all the experiments on 125M, 355M and 770M, we use the hyperparameters searched on the 30M model, which is $\gamma = 0.01$, $\beta_2 = 0.99$.

**Training Stability.** Sophia-H has better stability in pre-training compared to AdamW and Lion. Gradient clipping (by norm) is an important technique in language model pre-training as it avoids messing up the moment of gradients with one mini-batch gradient computed from rare data (Zhang et al., 2020). In practice, the frequency that gradients clipping is triggered is related to the training stability—if the gradient is frequently clipped, the iterate can be at a very instable state. We compare the proportion of steps where gradient clipping is triggered on GPT-2 small (125M) in Figure 6 (a). Although all methods use the same clipping threshold 1.0, Sophia-H seldomly triggers gradient clipping, while AdamW and Lion trigger gradient clipping in more than 10% of the steps.

One trick of pre-training deep Transformers is scaling the product of keys and values by the inverse of layer index as implemented by Mistral (Karamcheti et al., 2021) and Huggingface (Wolf et al., 2020). This stabilizes training and increases the largest possible learning rate. Without this trick, the maximum learning rate of AdamW and Lion on GPT-2 medium (355M) can only be 1.5e-4, which is much smaller than 3e-4 with the trick (the loss will blow up with 3e-4 without the trick). Moreover, the loss decreases much slower without the trick as shown in Figure 6 (b). In all the experiments, Sophia-H does not require scaling the product of keys and values by the inverse of the layer index.

## 3.5 Ablation Study

**Choices of Hessian update frequency** $k$. We study the effect of Hessian update frequency $k$ of Sophia-G on computational overhead and validation loss on a 30M GPT-2. We consider $k = 1, 10, 100$ and run each method for 100k, 200k, and 400k steps. All other hyperparameters are fixed, and we tune the peak learning rate with a grid search. We plot the amount of compute and the validation loss of each run in Figure 7 (a). While $k = 1$ has better validation loss with the same number of steps, the computational overhead is 50% and the speed w.r.t amount of compute is worse than $k = 10$. The choice of $k = 100$ still outperforms AdamW, but is not as good as $k = 10$.

**Diagonal Hessian pre-conditioners.** We compare different diagonal Hessian pre-conditioners (with the same $k = 10$ and $\gamma$ found by grid search): Empirical Fisher (E-F+clip), AdaHessian (AH+clip), Hutchinson (Sophia-H), and GNB (Sophia-G). Note that empirical Fisher is the EMA of squared gradients, which differs from GNB in label sampling. Results in Figure 7 (b) indicate that GNB is better than Empirical Fisher, which is consistent with Kunstner et al. (2019). Sophia-H is also consistently better than AdaHessian. We hypothesize that the difference stems from that the EMA of the diagonal Hessian estimates (used in Sophia-H ) has more denoising effect than the EMA of the second moment of Hessian estimates (used in AdaHessian).

**Element-wise Clipping.** We compare different update clipping strategies in Figure 7 (c): element-wise clipping without pre-conditioners (Clip), update normalization without pre-conditioners (Normalize), AdaHessian and Sophia-G without clipping (GNB). The learning rate is found by grid search. Note that clipping without pre-conditioner is essentially the same as sign momentum, or Lion with $\beta_1 = \beta_2$. Without element-wise clipping, we find that AdaHessian will diverge with $k = 2$ and GNB will diverge with $k = 5$, thus we use $k = 1$ for AdaHessian and $k = 2$ for GNB. Results indicate that per-coordinate clipping itself is already better than AdamW. Further adding the GNB pre-conditioner makes Sophia-G much better than baselines.

## 4 Conclusion

We introduced Sophia, a scalable second-order optimizer for language model pre-training. Sophia converges in fewer steps than first-order adaptive methods, while maintaining almost the same per-step cost. On language modeling with GPT models, Sophia achieves a 2x speed-up compared with AdamW in the number of steps, total compute, and wall-clock time.

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

## A    RELATED WORK

**Stochastic Adaptive First-order Optimizers in Deep Learning.** The idea of adaptive first-order optimizers dates back to RProp (Braun & Riedmiller, 1992). AdaGrad (Duchi et al., 2011) adapted the learning rate of features by estimated geometry and assign larger learning rate to infrequent features. RMSProp (Hinton et al., 2012) generalized RProp and is capable to work with smaller batch sizes. Adam (Kingma & Ba, 2014) improved RMSProp by introducing a running average of gradients, and has so far become the dominant approach to solve optimization problems in deep learning, especially for training Transformers (Vaswani et al., 2017). Many follow-up works proposed variants of Adam (Dozat, 2016; Shazeer & Stern, 2018; Reddi et al., 2019; Loshchilov & Hutter, 2017; Zhuang et al., 2020; You et al., 2019). Chen et al. (2023) performed a search over adaptive first-order algorithms and discovered Lion, which is a improved version of sign momentum SGD.

**Second-order Optimizers in Deep Learning.** Second-order optimizers are believed to have the potential to outperform adaptive first-order optimizers. Classical second-order optimization algorithms pre-condition the gradient with curvature information (Broyden, 1970; Nesterov & Polyak, 2006; Conn et al., 2000). Over the years, people have developed numerous ways to adapt these methods to deep learning. To the best of our knowledge, Becker & Le Cun (1988) was the first to use diagonal Hessian as the pre-conditioner. Martens et al. (2010) approximated the Hessian with conjugate gradient. Schaul et al. (2013) automatically tuned learning rate of SGD by considering diagonal Hessian. Pascanu & Bengio (2013) considered Gaussian Newton's approximation of Hessian and Fisher information matrix. Martens & Grosse (2015) and follow-up works (Ba et al., 2017; George et al., 2018; Martens et al., 2018; Zhang et al., 2022a) proposed to approximate the Hessian based on the structure of neural networks. Yao et al. (2021); Jahani et al. (2021) proposed to use the EMA of diagonal Hessian estimator as the pre-conditioner.

Despite these progress on deep learning applications, for decoder-only large language models, Adam still appears to the most popular optimizer. The authors of this paper suspect that many previous second-order optimizers face the challenge that the computational / memory overhead due to frequent Hessian computation hinders improvements in wall-clock time (Martens & Grosse, 2015; Gupta et al., 2018). Some of them also depend on specific model architecture or hardware structures, e.g., Anil et al. (2020) offloads hessian computation to CPUs, and George et al. (2018) needs ResNets and very large batch size to approximate the Fisher information matrix. To the best of our knowledge, there was no previous report that second-order optimizers can achieve a speed-up on large language models in total compute.

**Gradient Clipping.** Global gradient clipping has been a standard practice in pre-training language models (Merity et al., 2017; Radford et al., 2019; Izsak et al., 2021; Zhang et al., 2022b). It helps stabilizes training and avoids the effect of rare examples and large gradient noise. Zhang et al. (2019); Mai & Johansson (2021) showed that global gradient clipping is faster than standard SGD when global smoothness does not hold. Zhang et al. (2020); Crawshaw et al. (2022) found out per-coordinate gradient clipping can function as adaptivity. In addition to gradient clipping, Sophia is the first to clip the update (coordinate-wise) in second-order methods to avoid the effect of Hessian's changing along the trajectory and the inaccuracy of Hessian approximation.

**Optimization Algorithms in LM Pre-training.** Adam (Kingma & Ba, 2014) (with decoupled weight decay (Loshchilov & Hutter, 2017)) has become the dominant approach for language model pre-training (Vaswani et al., 2017; Devlin et al., 2018; Radford et al., 2019; Brown et al., 2020; Zhang et al., 2022b; Touvron et al., 2023). Different from vision tasks with CNNs (He et al., 2016) where models trained with SGD generalize better than models trained with Adam, Adam outperforms SGD by a huge margin on language modeling tasks with Transformers (Anil et al., 2019; Liu et al., 2020; Kunstner et al., 2023). Raffel et al. (2020); Chowdhery et al. (2022) trained Transformers with AdaFactor (Shazeer & Stern, 2018), which is a low rank version of Adam. You et al. (2019) proposed to make the update of Adam proportional to per-layer paramter norm to stably train LLMs.

# B    ADDITIONAL EXPERIMENT RESULTS

**Dynamics of Sophia in training.** We measure the $\ell_2$ norm of the EMA of the diagonal Hessian $h_t$, and the proportion of parameters where clipping happens (that is, $m_t/h_t$ is larger than $\gamma$) during pre-training in Figure 8. After the initial stage, the norm of the Hessian steadily grows. The proportion of parameters where clipping happens approaches 60%, which corroborates the importance of per-coordinate clipping in the algorithm.

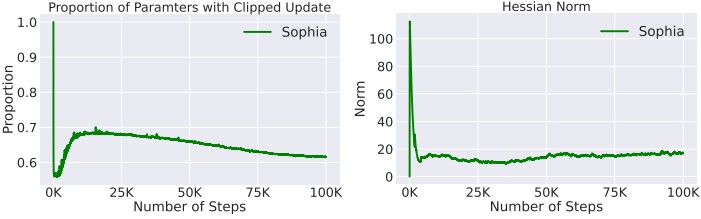

Figure 8: Visualization of training statistics. (a) The proportion of parameters whose update is clipped. (b) $\ell_2$ norm of the EMA of Hessian $h_t$.

**Results with different number of steps.** Due to space limit, runs with different number of steps and their comparison are provided in Figure 9. Across different total number of steps, Sophia outperforms AdamW and Lion with a large margin as the main experiments we presented in Section 3.3.

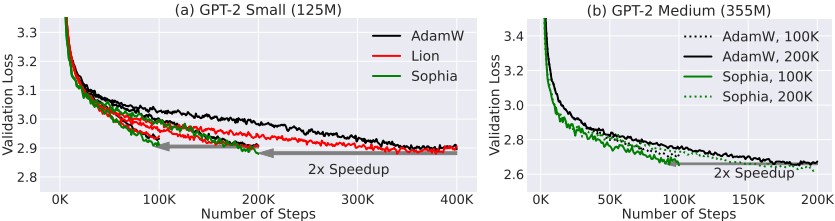

Figure 9: Results of training for different steps.

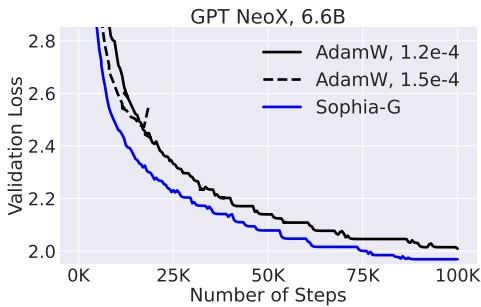

Figure 10: Results of 6.6B GPT NeoX tuning. We only carried two runs for AdamW and one run for Sophia on the 6.6B model. For AdamW, we tried peak learning rate 1.2e-4 (black solid curve) and 1.5e-4 (black dashed curve). The loss of the 1.5e-4 run blew up at about 17K steps. This indicates that 1.2e-4 is close to the largest peak learning rate for AdamW in that setup. Note that with the unstable 1.5e-4 learning rate, AdamW is slower than Sophia-G before blowing up.

# C    ADDITIONAL EXPERIMENT DETAILS

## C.1    HYPERPARAMTER TUNING

The hyperparameters we consider for baselines are as follows: peak learning rate, $\beta_1$, $\beta_2$ and weight decay. All hyperparameters except the peak learning rate are tuned with grid search on a 30M GPT-

Table 2: Model Configurations and Peak Learning Rate.

| Acronym | Size | d_model | n_head | depth | AdamW lr | Lion lr | Sophia-H lr | Sophia-G lr |
|---------|------|---------|--------|-------|----------|---------|-------------|-------------|
| –         | 30M  | 384  | 6  | 6  | 1.2e-3 | 4e-4   | 1e-3 | 1e-3   |
| Small     | 125M | 768  | 12 | 12 | 6e-4   | 1.5e-4 | 6e-4 | 6e-4   |
| Medium    | 355M | 1024 | 16 | 24 | 3e-4   | 6e-5   | 4e-4 | 4e-4   |
| –         | 540M | 1152 | 18 | 30 | 3e-4   | –      | 4e-4 | 4e-4   |
| Large     | 770M | 1280 | 20 | 36 | 2e-4   | –      | 3e-4 | 3e-4   |
| NeoX 1.5B | 1.5B | 1536 | 24 | 48 | 1.5e-4 | –      | –    | 1.2e-4 |
| NeoX 6.6B | 6.6B | 4096 | 32 | 32 | 1.2e-4 | –      | –    | 6e-5   |

2 trained for 50K steps. The peak learning rate is tuned on models of different sizes with grid search separately. We search $\beta_1$ in $[0.8, 0.9, 0.95, 0.96, 0.99]$ and $\beta_2$ in $[0.9, 0.95, 0.98, 0.99, 0.995]$. Weight decay is chosen from $0.05, 0.1, 0.2, 0.5$. For Lion, we also include $\beta_1 = 0.95, \beta_2 = 0.98$ as suggested by Chen et al. (2023). On 30M models, we found AdamW is sensitive to the choice of $\beta_1$ but not $\beta_2$. $\beta_1 = 0.9$ works the best for AdamW while $\beta_1 = 0.95$ works the best for Lion. We use $\beta_2 = 0.95$ for AdamW since this is the dominantly used configuration in the LLM pre-training literature, and $\beta_2 = 0.98$ for Lion as it is recommended by Chen et al. (2023). We found that weight decay 0.1 works the best for AdamW, while 0.2 works the best for Lion and Sophia.

For peak learning rate on 125M and 355M, we perform a grid search. For larger models, we search for the largest possible learning rate with which training does not blow up for each model in the following list: [6e-4, 4e-4, 3e-4, 2e-4, 1.5e-4, 1.2e-4, 1e-4, 8e-5, 6e-5, 4e-5]. For example, a 6.6B GPT NeoX model work with 1.2e-4 peak learning rate, but the loss will blow up if we increase the learning rate to 1.5e-4 as shown in Figure 10. The result of grid search of peak learning rate is provided in Table 2.

We use $\beta_1 = 0.96, \beta_2 = 0.99, \epsilon = $1e-12 and $k = 10$ for Sophia. We adopt the following procedure to obtain these hyper parameters. We first fix $\gamma = 0.01, k = 10$, and tune $\beta_1$ and $\beta_2$ with grid search on a 30M model, and directly use $\beta_1$ and $\beta_2$ from the 30M model on models of larger sizes. Similar to AdamW, we find that Sophia is not sensitive to $\beta_2$. We then fix $\beta_1 = 0.96, \beta_2 = 0.99$ and tuning $k = 10$. As shown in in Figure 7 (a), $k = 10$ is better than $k = 1$ or $k = 100$ in terms of the balance between convergence speed and the computation overhead.

After finding out $\beta_1 = 0.96, \beta_2 = 0.99, \epsilon = 1e-12$ and $k = 10$ with the method above, we tune $\gamma$ and peak learning rate jointly. We first tune $\gamma$ to make the proportion of coordinates where the update is not clipped (i.e., $|m_t / \max\{\gamma \cdot h_t, \epsilon\}| < 1$) in the range of $10\% - 50\%$. We search for $\gamma$ in the list of [0.005, 0.01, 0.02, 0.05, 0.1, 0.2]. As a result we find out $\gamma = 0.01$ works the best for Sophia-H while $\gamma = 0.05$ works the best for Sophia-G. We then fix $\beta_1, \beta_2 = 0.99, \epsilon, \gamma, k$ for all larger models.

To tune the peak learning rate, we adopt the same procedure as we use for baseline methods. The result of grid search of peak learning rate is also provided in Table 2.

## C.2 MODEL AND IMPLEMENTATION DETAILS

We consider three sizes of GPT-2 corresponding to small, medium, and large in Radford et al. (2019). We also introduce a 30M model for efficient hyperparameter grid search and a 540M model for scaling law visualization. We provide the model specifications in Table 2. We use the nanoGPT (https://github.com/karpathy/nanoGPT/) code base. Following nanoGPT, we use GELU activations and disable bias and Dropout Srivastava et al. (2014) during pre-training.

GPT-2 models are trained on OpenWebText (Gokaslan & Cohen, 2019). The text is tokenized with the GPT-2 tokenizer (Radford et al., 2019). We use the train and validation split from nanoGPT. The training set contains 9B tokens, and the validation set contains 4.4M tokens.

We use distributed data parallel with gradient accumulation to enable a batch size of 480. All models are trained with bfloat16. The 125M and 355M models are trained on machines with 10 A5000 GPUs, while the 770M models are trained on an AWS p4d.24xlarge instance with 8 A100 GPUs.

We consider 1.5B and 6.6B GPT NeoX (Black et al., 2022) models trained on the Pile (Gao et al., 2020). The models use GPT NeoX tokenizers. We use levanter (https://github.com/stanford-crfm/levanter/tree/main) for GPT NeoX. We use fully sharded data parallel with gradient accumulation to enable a batch size of 512 for the 1.5B model and 1024 for the 6.6B model. These models are trained on a TPU v3-128 slice.

The context is a passages containing some information. Given a question about the context, use the information to answer the question with either 'Yes' or 'No'.

Context: 3-way lamp -- The center contact of the bulb typically connects to the medium-power filament, and the ring connects to the low-power filament. Thus, if a 3-way bulb is screwed into a standard light socket that has only a center contact, only the medium-power filament operates. In the case of the 50 W / 100 W / 150 W bulb, putting this bulb in a regular lamp socket will result in it behaving like a normal 100W bulb. Question: do 3-way light bulbs work in any lamp
Answer: Yes

Context: Perfume: The Story of a Murderer (film) -- Perfume: The Story of a Murderer is a 2006 German period psychological crime thriller film directed by Tom Tykwer and starring Ben Whishaw, Alan Rickman, Rachel Hurd-Wood, and Dustin Hoffman. Tykwer, with Johnny Klimek and Reinhold Heil, also composed the music. The screenplay by Tykwer, Andrew Birkin, and Bernd Eichinger is based on Patrick Süskind's 1985 novel Perfume. Set in 18th century France, the film tells the story of Jean-Baptiste Grenouille (Whishaw), an olfactory genius, and his homicidal quest for the perfect scent. Question: is the film perfume based on a true story
Answer: No

BoolQ

Given a premise and a hypothesis, answer whether the hypothesis follows from the premise with 'Yes' or 'No'.

Context: The Bank of Italy, the ultimate arbiter of Italian banking mergers, has been engulfed by scandal since police wire taps revealed Fazio and his wife advised a local banker in a bid for Bank Antonveneta against Dutch bank ABN AMRO.
Question: A local banker bids for Bank Antonveneta.
Answer: Yes

Context: The Statue of Liberty was reopened to the public on July 5 after its extensive refurbishing. 1986 is a common year starting on Wednesday of the Gregorian calendar.
Question: The Statute of Liberty was built in 1986.
Answer: None

RTE

Given a premise and a hypothesis, answer whether the hypothesis logically follows from the premise with 'True' or 'False' or 'Neither'.

Context: B: She says that when her husband died oh, that my uncle had said that he would never put her in a rest home. So it's kind of, uh, I don't know. I mean, I don't think my parents would but she is getting pretty bad like she has to have like a little toilet right by her bed and, it's, A: Uh-huh. B: and my mom has to take care of her pretty much so it gets, I don't know. it's a hard decision, but I don't think I would do it to my parents personally. Question: she would do it to her parents
Answer: No

Context: B: No, it was, I didn't like the way it ended. A: I know, well the only reason I know whxy it ended is on Arsenio Hall one night, Christopher Reeves told, that, you know, B: Uh-huh. A: I can't believe they killed them. Question: they killed them
Answer: Yes

CB

Choose the correct ending for the context.

Choice1: the woman kissed him.
Choice2: the woman made him blush.
Context: The man had lipstick on his cheek because
Answer: Choice1

Choice1: i attended a yoga class.
Choice2: i bought fruits and vegetables.
Context: I made a resolution to eat a healthy diet so
Answer: Choice2

COPA

Figure 11: Prompts for SuperGLUE downstream evaluation.

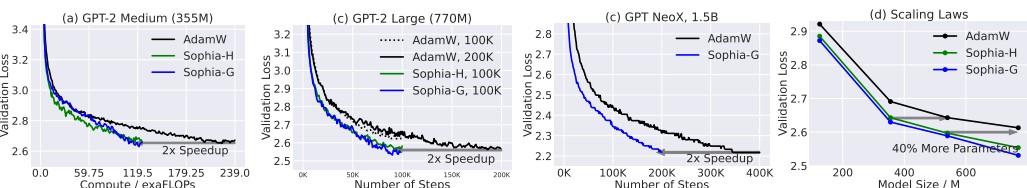

Figure 12: An enlarged version of Figure 1.

We observed AdamW and Lion does not perform well on standard transformers which are larger than 355M. The iterates become unstable when the learning rate is close to the choice of Radford et al. (2019). We introduce scaling attention by the inverse of layer index to address this issue following Karamcheti et al. (2021); Wolf et al. (2020). Note that Sophia does not need this trick as mentioned in Section 3.4.

## C.3  DOWNTREAM EVALUATION

We perform few-shot evaluation of the models on 4 subtasks of SuperGLUE. We use 2-shot prompting and greedy decoding. The prompt consists of an instruction followed by two examples. The examples are sampled from the train split while we report the accuracy on validation split averaged over 5 selection of exemplars. Prompts for each subtask are illustrated in Figure 11.

## D  A DETAILED EXPLANATION OF GNB ESTIMATOR

We leverage the structure of the loss to design a biased stochastic estimator for the diagonal Hessian, following Schraudolph (2002); Martens (2020); Wei et al. (2020). Suppose $\ell(\theta, (x, y))$ is a loss function on an example $(x, y)$ of the form $\ell(\theta, (x, y)) = \ell_{ce}(f(\theta, x), y)$ where $\ell_{ce}$ is the cross-entropy loss and $f(\theta, x) \in \mathbb{R}^V$ is the logits, and $V$ is the number of items/classes in a multi-class classification problem (e.g., the vocabulary size in LLMs). First, the Hessian of $\ell(\theta, (x, y))$ (w.r.t to variable $\theta$) has the well-known Gauss-Newton decomposition (Ortega & Rheinboldt, 2000; Schraudolph, 2002) (which is a simple consequence of the chain rule),

$$\nabla_\theta^2 \ell(\theta) = J_\theta f(\theta, x) \cdot S \cdot J_\theta f(\theta, x)^\top + J_{\theta\theta} f(\theta, x)[q] \tag{11}$$

where $J_\theta f(\theta, x)$ is the Jacobian of $f$ w.r.t to $\theta$ viewed as a matrix in $\mathbb{R}^{d \times V}$, $S = \frac{\partial^2 \ell_{ce}(t, y)}{\partial t^2}\Big|_{t=f(\theta, x)} \in \mathbb{R}^{V \times V}$ is the second-order derivatives of the loss w.r.t to the logits, $q = \frac{\partial \ell_{ce}(t, y)}{\partial t}\Big|_{t=f(\theta, x)} \in \mathbb{R}^V$ is the first-order derivatives of the loss w.r.t to the logits, and $J_{\theta\theta} f(\theta, x)$ is the second-order derivatives of the multi-variate function $f(\theta, x)$ w.r.t $\theta$, viewed as a linear map from $\mathbb{R}^V$ to $\mathbb{R}^{d \times d}$, where $d$ is the dimension of the parameter $\theta$.

In the context of neural networks, past works have found that the second term $J_{\theta\theta} f(\theta, x)[q]$ in Equation 11 is often relative smaller than the first term $J_\theta f(\theta, x) \cdot S \cdot J_\theta f(\theta, x)^\top$ (Sankar et al., 2021), which is often referred to as the Gauss-Newton matrix (Dennis Jr & Schnabel, 1996; Ortega & Rheinboldt, 2000; Schraudolph, 2002; Chen, 2011) and used as pre-conditioners in second-order optimizers (Botev et al., 2017; Martens, 2020; Gargiani et al., 2020). Following this line of work, we build an unbiased estimator for *the diagonal* of the Gauss-Newton matrix, which is a biased estimator for the diagonal of the Hessian.

We first claim that $S$ only depends $f(\theta, x)$ but not $y$, even though the loss depends on $y$.[3] Thus, $S = \frac{\partial^2 \ell_{ce}(t, \hat{y})}{\partial t^2}\Big|_{t=f(\theta, x)}$ for any $\hat{y} \in \{1, \ldots, V\}$, which implies that $S = \mathbb{E}_{\hat{y} \sim p(\theta, x)}\left[\frac{\partial^2 \ell_{ce}(t, \hat{y})}{\partial t^2}\Big|_{t=f(\theta, x)}\right]$.
Because $\ell_{ce}(t, y)$ is the negative log-probability of the probabilistic model defined by the categorical distribution $\mathrm{Cat}(t)$ with parameter $t$, by Bartlett's second identity (Bartlett, 1953), we have that,

$$S = \mathbb{E}_{\hat{y} \sim \mathrm{Cat}(t)}\left[\frac{\partial^2 \ell_{ce}(t, \hat{y})}{\partial t^2}\right] = \mathbb{E}_{\hat{y} \sim \mathrm{Cat}(t)}\left[\frac{\partial \ell_{ce}(t, \hat{y})}{\partial t}\frac{\partial \ell_{ce}(t, \hat{y})}{\partial t}^\top\right], \tag{12}$$

where the first equality holds for $t = f(\theta, x)$ and the second equality holds for all $t$ by Bartlett's second identity. Therefore, the Gauss-Newton matrix satisfies

$$J_\theta f(\theta, x) \cdot S \cdot J_\theta f(\theta, x)^\top = \mathbb{E}_{\hat{y} \sim \mathrm{Cat}(t)}\left[J_\theta f(\theta, x)\frac{\partial \ell_{ce}(t, \hat{y})}{\partial t}\frac{\partial \ell_{ce}(t, \hat{y})}{\partial t}^\top J_\theta f(\theta, x)^\top\right]$$

$$= \mathbb{E}_{\hat{y} \sim \mathrm{Cat}(t)}\left[\nabla_\theta \ell_{ce}(f(\theta, x), \hat{y})\nabla_\theta \ell_{ce}(f(\theta, x), \hat{y})^\top\right], \tag{13}$$

which implies that $\mathrm{diag}(J_\theta f(\theta, x) \cdot S \cdot J_\theta f(\theta, x)^\top) = \mathbb{E}_{\hat{y} \sim \mathrm{Cat}(t)}[\nabla_\theta \ell_{ce}(f(\theta, x), \hat{y}) \odot \nabla_\theta \ell_{ce}(f(\theta, x), \hat{y})]$. Hence, the quantity $\ell_{ce}(f(\theta, x), \hat{y}) \odot \nabla_\theta \ell_{ce}(f(\theta, x), \hat{y})$ is an unbiased estimator of the Gauss-Newton matrix for the Hessian of a one-example loss $\ell(f(\theta, x), y)$.

*Mini-batch version.* Given a mini-batch of inputs $\{(x_b, y_b)\}_{b=1}^B$. The most natural way to build an estimator for the diagonal of the Gauss-Newton matrix for the Hessian of the mini-batch loss is using

$$\frac{1}{B}\sum_{b=1}^B \nabla \ell_{ce}(f(\theta, x_b), \hat{y}_b) \odot \nabla_\theta \ell_{ce}(f(\theta, x_b), \hat{y}_b), \tag{14}$$

---

[3]Denote by $p(\theta, x) = \mathrm{softmax}(f(\theta, x)) \in \mathbb{R}^V$ the probability vector obtained by applying softmax on the logits. Indeed, a simple derivation shows that $S = \mathrm{diagonal}(p(\theta, x)) - p(\theta, x)p(\theta, x)^\top$, where $\mathrm{diagonal}(p(\theta, x))$ is the matrix with the vector $p(\theta, x)$ residing on the diagonal. In fact, this is a general property of exponential families—the Hessian of the negative log-likelihood of any exponential family distribution only depends on the parameters of that exponential family, but not on the example on which the likelihood is evaluated.

where $\hat{y}_b$'s are labels sampled from the model on inputs $x_b$'s respectively. However, as noted by Grosse (2022), implementing this estimator is inconvenient under the current auto-differentiation frameworks, where the users only have access to the average gradient over a mini-batch (as opposed to the individual ones). Fortunately, by the Bartlett's first identity (Bartlett, 1953) (which generally holds for the negative log-likelihood loss of any probabilistic model), we have:

$$\forall b, \quad \mathbb{E}_{\hat{y}_b} \nabla \ell_{\text{ce}}(f(\theta, x_b), \hat{y}_b) = 0. \tag{15}$$

Let $\widehat{L}(\theta) = \frac{1}{B} \sum_{b=1}^{B} \ell_{\text{ce}}(f(\theta, x_b), \hat{y}_b)$ be the mini-batch loss on the *sampled* labels (as opposed to the original labels). Observing that $\hat{y}_b$'s are independent with each other, we have

$$\mathbb{E}_{\hat{y}'_b s} \left[ B \cdot \nabla_\theta \widehat{L}(\theta) \odot \nabla_\theta \widehat{L}(\theta) \right] = \mathbb{E}_{\hat{y}'_b s} \left[ \frac{1}{B} \sum_{b=1}^{B} \nabla \ell_{\text{ce}}(f(\theta, x_b), \hat{y}_b) \odot \sum_{b=1}^{B} \nabla \ell_{\text{ce}}(f(\theta, x_b), \hat{y}_b) \right]$$

$$= \mathbb{E}_{\hat{y}'_b s} \left[ \frac{1}{B} \sum_{b=1}^{B} \nabla \ell_{\text{ce}}(f(\theta, x_b), \hat{y}_b) \odot \nabla \ell_{\text{ce}}(f(\theta, x_b), \hat{y}_b) \right]. \tag{16}$$

Note that the RHS of Equation 16 is the same as the expectation of Equation 14, which, by Equation 13, also equals to the diagonal of the Gauss-Newton matrix for the mini-batch loss. Hence, we use $B \cdot \nabla_\theta \widehat{L}(\theta) \odot \nabla_\theta \widehat{L}(\theta)$ as the estimator.

To the best of our knowledge, this estimator of Gauss-Newton matrix was first used in Wei et al. (2020). Given the use Bartlett's first and second identities that are central to the estimator, we call it Gauss-Newton-Bartlett (GNB) estimator.

# E    LIMITATIONS

**Scaling up to larger models and datasets.** Although Sophia demonstrates scalability up to 770M models and OpenWebText, and there is no essential constraints from further scaling up, we do not compare with AdamW and Lion on larger models and datasets due to limited resources. We believe Sophia is faster than AdamW and Lion on larger models given the improvement in scaling laws and better pre-training stability.

**Holistic downstream evaluation.** We evaluate pre-trained checkpoints on 4 SuperGLUE subtasks, which only demonstrates the improvement in downstream performance several datasets. While a holistic evaluation of language models itself is an open research topic, better downstream evaluation is still important. The limitation in downstream evaluation is also due to the limited model size, because language models at this scale do not have enough capabilities such as in-context learning, and mathematical reasoning.

**Evaluation on other domains.** While this paper focuses on optimizers for large language modeling, a more general optimizer should also be evaluated in other domains such as computer vision, reinforcement learning, and multimodel tasks. Due to the limitation of computation resources, we leave the application to other domains and models to future works.

# F    THEORETICAL ANALYSIS

This section provides runtime bounds for the deterministic version of Sophia that does not depend on the local condition number (the ratio between maximum and minimum curvature at the local minimum) and the worst-case curvature (that is, the smoothness parameter), demonstrating the advantage of Sophia in adapting to heterogeneous curvatures across parameter dimensions.

We start with standard assumptions on the differentiability and uniqueness of the minimizer.

**Assumption F.1.** $L : \mathbb{R}^d \to \mathbb{R}$ *is a twice continuously differentiable, strictly convex function with* $\theta^*$ *being its minimizer. For convenience, we denote* $\lambda_{\min}(\nabla^2 L(\theta^*))$ *by* $\mu$.

The following assumptions state that the Hessian has a certain form of continuity—within a neighborhood of size $R$, the ratio between the Hessians, $\nabla^2 L(\theta')^{-1} \nabla^2 L(\theta)$, is assumed to be bounded by a constant 2.

**Assumption F.2.** *There exists a constant* $R > 0$*, such that*

$$\forall \theta, \theta' \in \mathbb{R}^d, \|\theta - \theta'\|_2 \leq R \implies \left\| \nabla^2 L(\theta')^{-1} \nabla^2 L(\theta) \right\|_2 \leq 2 \tag{17}$$

We analyze the convergence rate of the deterministic version of the Sophia on convex functions,

$$\theta_{t+1} = \theta_t - \eta V_t^\top \text{clip}(V_t(\nabla^2 L(\theta_t))^{-1}\nabla L(\theta_t), \rho), \tag{18}$$

where $\nabla^2 L(\theta_t) = V_t^\top \Sigma_t V_t$ is an eigendecomposition of $\nabla^2 L(\theta_t)$. Here, we use the full Hessian as the pre-conditioner because the diagonal Hessian pre-conditioner cannot always work for general functions which may not have any alignment with the natural coordinate system. Moreover, the matrix $V_t$ transforms $(\nabla^2 L(\theta_t))^{-1}\nabla L(\theta_t)$ into eigenspace and thus the clipping can be done element-wise in the eigenspace. We do not need the max between Hessian and $\epsilon$ in the original version of Sophia because the Hessian is always PSD for convex functions. Finally, the matrix $V_t^\top$ transforms the update back to the original coordinate system for the parameter update.

**Theorem F.3.** *Under Assumption F.1 and Assumption F.2, let $\eta = 1/2, \rho = \frac{R}{2\sqrt{d}}$, the update in Equation 18 reaches a loss at most $\epsilon$ in $T \lesssim d \cdot \frac{L(\theta_0) - \min L}{\mu R^2} + \ln \frac{\mu R^2}{32 d \epsilon}$ steps.*

The first term in the runtime bound is a burn-in time before reaching a local region, where the error decays exponentially fast so that the runtime bound is logarithmic in $1/\epsilon$ as the second term in the runtime bound shows. We remark that the bound does not depend on the condition number (the ratio between the maximum and minimum eigenvalue of Hessian), as opposed to the typical dependency on the maximum eigenvalue of the Hessian (or the smoothness parameter) in standard analysis of gradient descent in convex optimization (Boyd & Vandenberghe, 2004). Moreover, even on simple quadratic functions, the convergence rate of simplified Adam (SignGD) depends on the condition number (Appendix G.1). This demonstrates the advantage of Sophia in adapting to heterogeneous curvatures across parameter dimensions.

## G  THEORETICAL ANALYSES: DETAILS OF SECTION F

Theorem F.3 is a direct combination of the Lemma G.10 (Descent Lemma), Lemma G.9 and Lemma G.11. In the analysis, there will be two phases. In the first phase decrease loss to $\frac{\mu\rho^2}{8}$ in $8\frac{L(\theta(0)) - \min L}{\eta\mu\rho^2}$ steps. In the second phase, there will be an exponential decay of error.

**Lemma G.1.** *Under Assumption F.1, we have that $L(\theta) \to \infty$ whenever $\|\theta\|_2 \to \infty$.*

*Proof of Lemma G.1.* By convexity of $L$, we have $\forall \theta \in \mathbb{R}^d$ with $\|\theta - \theta^*\|_2 \geq 1$,

$$\frac{1}{\|\theta - \theta^*\|_2}L(\theta) + \frac{\|\theta - \theta^*\|_2 - 1}{\|\theta - \theta^*\|_2}L(\theta^*) \geq L(\theta^* + \frac{\theta - \theta^*}{\|\theta - \theta^*\|_2}) \geq \min_{\|\bar{\theta}\|_2 = 1} L(\theta^* + \bar{\theta}). \tag{19}$$

Since $L$ is strictly convex, $\Delta \triangleq \min_{\|\bar{\theta}\|_2 = 1} L(\theta^* + \bar{\theta}) - L(\theta^*) > 0$. Thus we conclude that

$$L(\theta) \geq \|\theta - \theta^*\|_2 \Delta + L(\theta^*) \geq (\|\theta\|_2 - \|\theta^*\|_2)\Delta + L(\theta^*). \tag{20}$$

Therefore when $\|\theta\|_2 \to \infty$, $L(\theta) \to \infty$ as well. $\square$

Note that we don't assume the Hessian of loss is Lipschitz. Assumption F.2 only assumes the Hessian in a neighborhood of constant radius only differs by a constant in the multiplicative sense.

**Lemma G.2.** *For any $\theta \in \mathbb{R}^d$ satisfying $L(\theta) - \min L \leq \frac{\mu R^2}{4}$, it holds that $\|\theta - \theta^*\|_2 \leq 2\sqrt{\frac{L(\theta) - \min L}{\mu}} \leq R$.*

*Proof of Lemma G.2.* We will prove by contradiction. Suppose there exists such $\theta$ with $L(\theta) \leq \frac{\mu R^2}{4}$ but $\|\theta - \theta^*\|_2 > 2\sqrt{\frac{L(\theta) - \min L}{\mu}}$. We consider $\theta' \triangleq \theta^* + \sqrt{\frac{2L(\theta)}{\mu}} \cdot \frac{\theta - \theta^*}{\|\theta - \theta^*\|_2}$. Since $\theta'$ is between $\theta$ and $\theta^*$ and that $L$ is strictly convex, we know that $L(\theta') < L(\theta)$. However, by Taylor expansion on function $f(t) \triangleq L(\theta^* + t(\theta' - \theta^*))$, we have that

$$f(1) = f(0) + f'(0) + \frac{f''(t)}{2}, \quad \text{for some } t \in [0, 1]. \tag{21}$$

Note that $\|\theta' - \theta^*\|_2 \le \|\theta - \theta^*\|_2 \le R$, by Assumption F.2 and Assumption F.1, we have $f''(t) = (\theta' - \theta^*)^\top \nabla^2 L(t\theta' + (1-t)\theta^*)(\theta' - \theta^*) \ge \frac{1}{2}(\theta' - \theta^*)^\top \nabla^2 L(\theta^*)(\theta' - \theta^*) \ge \frac{\mu}{2}\|\theta' - \theta^*\|_2^2 = 2(L(\theta) - \min L))$. Also note that $f(1) = L(\theta')$, $f(0) = L(\theta^*)$ and $f'(0) = 0$, we conclude that $L(\theta') - L(\theta^*) \ge L(\theta) - L(\theta^*)$, namely $(\theta') \ge L(\theta)$. Contradiction! $\square$

**Lemma G.3.** *For any $\theta \in \mathbb{R}^d$ satisfying that $\|\nabla L(\theta)\|_2 \le \frac{R\mu}{2}$, it holds that $\|\theta - \theta^*\|_2 \le \frac{2\|\nabla L(\theta)\|}{\mu} \le R$.*

*Proof of Lemma G.3.* We will prove by contradiction. We consider function $f(t) \triangleq \left\langle \frac{\theta - \theta^*}{\|\theta - \theta^*\|_2}, \nabla L(\theta^* + t \cdot \frac{\theta - \theta^*}{\|\theta - \theta^*\|_2}) \right\rangle$. Because of the strict convexity of $L$, $f$ is a strict monotone increasing function. If $\|\theta - \theta^*\| > \frac{2\|\nabla L(\theta)\|}{\mu}$ but $\|\nabla L(\theta)\|_2 \le \frac{R\mu}{2}$, then we have $f(R) < f(\|\theta - \theta^*\|_2) \le \|\nabla L(\theta)\|_2$. On the other hand, by Assumption F.2 and Assumption F.1, $f'(t) \ge \frac{\mu}{2}$ for $t \in [0, R]$. Thus $f(R) \ge f(0) + \int_{t=0}^{\frac{2\|\nabla L(\theta)\|}{\mu}} f'(t)\mathrm{d}t = \|\nabla L(\theta)\|$. Contradiction! $\square$

**Lemma G.4.** *For any $\theta \in \mathbb{R}^d$, the following differential equation has at least one solution on interval $[0, 1]$:*

$$\frac{\mathrm{d}\theta(t)}{\mathrm{d}t} = -(\nabla^2 L(\theta(t)))^{-1}\nabla L(\theta), \quad \theta(0) = \theta, \tag{22}$$

*and the solution satisfies that $\nabla L(\theta(t)) = (1-t)\nabla L(\theta)$ for all $t \in [0,1]$ and $\theta(0) = \theta^*$.*

*Proof of Lemma G.4.* Since $\nabla^2 L$ is continuous and positive definite by Assumption F.1 , $(\nabla^2 L)^{-1}$ is continuous and thus the above ODE (47) has a solution over interval $[0, T)$ for some positive $T$ and we let $T_{\max}$ be the largest positive number such that the solution exists (or $T_{\max} = \infty$). Now we claim $T_{\max} \ge 1$, otherwise $\|\theta(t) - \theta^*\|_2$ must diverge to infinity when $t \to T_{\max}$. However, for any $t \le 1$, we have

$$\frac{\mathrm{d}\nabla L(\theta(t))}{\mathrm{d}t} = -\nabla L(\theta), \tag{23}$$

which implies that $\nabla L(\theta(t)) = (1-t)\nabla L(\theta)$ for all $t \in [0, 1]$. Therefore,

$$\frac{\mathrm{d}L(\theta(t))}{\mathrm{d}t} = -(\nabla L(\theta(t)))^\top (\nabla^2 L(\theta(t)))^{-1}\nabla L(\theta) = (1-t)(\nabla L(\theta))^\top (\nabla^2 L(\theta(t)))^{-1}\nabla L(\theta) \le 0. \tag{24}$$

Thus $L(\theta(t)) \le L(\theta(0))$. By Lemma G.1, we know that $\|\theta(t)\|$ remains bounded for all $t \in [0, T_{\max}]$, thus $T_{\max} \ge 1$. Note that $\theta(1)$ has zero gradient, $\theta(1)$ must be $\theta^*$. This completes the proof. $\square$

**Lemma G.5.** *For any $\theta \in \mathbb{R}^d$ satisfying (1) $L(\theta) - \min L \le \frac{\mu R^2}{16}$ or (2) $\|\nabla L(\theta)\|_2 \le \frac{R\mu}{4}$, it holds that*

$$L(\theta) - \min L \le \nabla L(\theta)^\top (\nabla^2 L(\theta))^{-1}\nabla L(\theta) \le 4(L(\theta) - \min L). \tag{25}$$

*Proof of Lemma G.5.* Let $\{\theta(t)\}_{t=0}^1$ be the solution of Equation 47. We know that $\nabla L(\theta(t)) = (1-t)\nabla L(\theta)$ for all $t \in [0, 1]$ and that $\theta(1) = \theta^*$ by Lemma G.4. For case (1), by Lemma G.2, we know that for any $t \in [0, 1]$, $\|\theta(t) - \theta^*\|_2 \le R/2$. For case (2), by Lemma G.3, we know that for any $t \in [0, 1]$, $\|\theta(t) - \theta^*\|_2 \le R/2$. Thus in both two cases, $\|\theta(t) - \theta\|_2 = \|\theta(t) - \theta(0)\|_2 = \le \|\theta(t) - \theta^*\| + \|\theta(0) - \theta^*\| \le R$. By Assumption F.2, it holds that

$$2(\nabla^2 L(\theta))^{-1} \succeq (\nabla^2 L(\theta(t)))^{-1} \succeq \frac{1}{2}(\nabla^2 L(\theta))^{-1}. \tag{26}$$

for all $t \in [0, 1]$. Therefore, we have that

$$L(\theta) - \min L = L(\theta(0)) - L(\theta(1)) = \int_{t=0}^1 (\nabla L(\theta(t)))^\top (\nabla^2 L(\theta(t)))^{-1}\nabla L(\theta)$$

$$= \int_{t=0}^1 (1-t)(\nabla L(\theta))^\top (\nabla^2 L(\theta(t)))^{-1}\nabla L(\theta). \tag{27}$$

The proof is completed by plugging Equation 26 into Equation 27 and noting that $\int_{t=0}^1 (1-t) = 1/2$. $\square$

**Lemma G.6.** *For any $\theta \in \mathbb{R}^d$ satisfying (1) $L(\theta) - \min L \le \frac{\mu R^2}{4}$ or (2) $\|\nabla L(\theta)\|_2 \le \frac{R\mu}{2}$, it holds that*

$$L(\theta) - \min L \le \mu^{-1} \|\nabla L(\theta)\|_2^2 \tag{28}$$

*Proof of Lemma G.6.* The proof of Lemma G.6 is almost the same as that of Lemma G.5 and thus omitted. $\square$

**Lemma G.7.** *For any $\theta \in \mathbb{R}^d$ satisfying $L(\theta) - \min L \le \frac{\mu R^2}{16}$, it holds that*

$$\left\|(\nabla^2 L(\theta))^{-1}\nabla L(\theta)\right\|_2 \le \sqrt{\frac{8(L(\theta) - \min L)}{\mu}}. \tag{29}$$

*Proof of Lemma G.7.* By Lemma G.2, we have that $\|\theta - \theta^*\|_2 \le R$. By Assumption F.2, we have $\nabla^2 L(\theta) \succeq \frac{1}{2}\nabla^2 L(\theta^*) \succeq \frac{\mu}{2}I_d$. By Lemma G.5, we have that

$$4(L(\theta) - \min L) \ge \nabla L(\theta)^\top (\nabla^2 L(\theta))^{-1} \nabla L(\theta) \tag{30}$$

$$\ge \nabla L(\theta)^\top (\nabla^2 L(\theta))^{-1} \nabla^2 L(\theta)(\nabla^2 L(\theta))^{-1} \nabla L(\theta) \tag{31}$$

$$\ge \frac{\mu}{2} \left\|\nabla L(\theta)^\top (\nabla^2 L(\theta))^{-1}\right\|_2^2. \tag{32}$$

This completes the proof. $\square$

**Lemma G.8.** *For any $\theta \in \mathbb{R}^d$ satisfying that $\left\|((\nabla^2 L(\theta))^{-1}\nabla L(\theta)\right\|_2 \le \frac{R}{2}$, it holds that*

$$L(\theta) - \min L \le \nabla L(\theta)^\top (\nabla^2 L(\theta))^{-1} \nabla L(\theta) \le 4(L(\theta) - \min L). \tag{33}$$

*Proof of Lemma G.8.* Let $\{\theta(t)\}_{t=0}^1$ be the solution of Equation 47 and we claim that for all $t \in [0,1]$, $\|\theta(t) - \theta\|_2 \le R$. Otherwise, let $T$ be the smallest positive number such that $\|\theta(T) - \theta\|_2 = R$. Such $T$ exists because $\|\theta(t) - \theta\|_2$ is continuous in $t$ and $\|\theta(0) - \theta\|_2 = 0$. We have that

$$R = \|\theta(T) - \theta(0)\|_2 \le \int_{t=0}^T \left\|\frac{\mathrm{d}\theta(t)}{\mathrm{d}t}\right\|_2 \mathrm{d}t \tag{34}$$

$$= \int_{t=0}^T \left\|((\nabla^2 L(\theta(t)))^{-1}\nabla L(\theta)\right\|_2 \mathrm{d}t \tag{35}$$

$$\le \int_{t=0}^T \left\|(\nabla^2 L(\theta(t)))^{-1}\nabla^2 L(\theta)\right\|_2 \left\|((\nabla^2 L(\theta))^{-1}\nabla L(\theta)\right\|_2 \mathrm{d}t \tag{36}$$

$$\le 2\int_{t=0}^T \left\|((\nabla^2 L(\theta))^{-1}\nabla L(\theta)\right\|_2 \mathrm{d}t \tag{37}$$

$$\le 2T\frac{R}{2} = RT, \tag{38}$$

which implies $T = 1$. Here in Equation 37, we use Assumption F.2. Thus we conclude that for all $t \in [0,1]$, $\|\theta(t) - \theta\|_2 \le R$. By Assumption F.2, it holds that

$$2(\nabla^2 L(\theta))^{-1} \succeq (\nabla^2 L(\theta(t)))^{-1} \succeq \frac{1}{2}(\nabla^2 L(\theta))^{-1}. \tag{39}$$

Therefore, we have that

$$L(\theta) - \min L = L(\theta(0)) - L(\theta(1)) = \int_{t=0}^1 (\nabla L(\theta(t)))^\top (\nabla^2 L(\theta(t)))^{-1} \nabla L(\theta)$$

$$= \int_{t=0}^1 (1-t)(\nabla L(\theta))^\top (\nabla^2 L(\theta(t)))^{-1} \nabla L(\theta). \tag{40}$$

The proof is completed by plugging Equation 39 into Equation 40 and noting that $\int_{t=0}^1 (1-t) = 1/2$. $\square$

**Lemma G.9.** *If $\rho \leq \frac{R}{2\sqrt{d}}$, then for any $\Delta \leq \frac{R\rho\mu}{10}$ and any $\theta \in \mathbb{R}^d$ satisfying*

$$\sum_{i=1}^{d} \min\{\rho \left|v_i^\top \nabla L(\theta)\right|, \sigma_i^{-1} \left|v_i^\top \nabla L(\theta)\right|^2\} \leq \Delta, \tag{41}$$

*where $\nabla^2 L(\theta) = V^\top \Sigma V$ is the eigendecomposition of $\nabla^2 L(\theta)$, $v_i$ is the $i$th row of $V$ and $\Sigma = \mathrm{diag}(\sigma_1, \ldots, \sigma_d)$, it holds that*

$$L(\theta) - \min L \leq \Delta + \frac{25\Delta^2}{\rho^2 \mu} \tag{42}$$

*In particular, if we set $\Delta \triangleq \frac{\mu\rho^2}{20}$, we have $L(\theta) - \min L \leq \frac{\mu\rho^2}{8}$.*

*Proof of Lemma G.9.* Let $I_\theta \triangleq \{i \in [d] \mid \left|v_i^\top \nabla L(\theta)\right| \sigma_i^{-1} \leq \rho\}$ be the set of indices where clipping does not happen. Then we have that

$$\sum_{i \in I_\theta} \sigma_i^{-1} \left|v_i^\top \nabla L(\theta)\right|^2 \leq \Delta \tag{43}$$

$$\sum_{i \notin I_\theta} \rho \left|v_i^\top \nabla L(\theta)\right| \leq \Delta \tag{44}$$

Now we consider a new strictly convex loss function in $R^{|I_\theta|}$, which is $L$ restricted on the space of $\{\theta + \sum_{i \in I_\theta} w_{[i]} v_i \mid w \in \mathbb{R}^{|I_\theta|}\}$, that is, $L_\theta(w) = L(\theta + \sum_{i \in I_\theta} w_{[i]} v_i)$. This new loss function $L_\theta$ clearly satisfy Assumption F.2 since it is a restriction of $L$ into some subspace of $\mathbb{R}^d$. By Lemma G.1, we know that $\inf_w L_\theta(w)$ can be attained and we denote it by $w^*$. By Assumption F.1, we know that $L_\theta$ is strictly convex and thus $\nabla^2 L_\theta(w) \succ 0$, which means Assumption F.1 also holds for $L_\theta$.

Next we will apply Lemma G.8 on $L_\theta$ at $w = 0$. We use $V_{I_\theta} \in \mathbb{R}^{|I| \times d}$ to denote the submatrix of $V$ containing rows in $I$ for any $I \subset [d]$. One can verify by chain rule that $\nabla L_\theta(w) = V_{I_\theta} \nabla L(\theta + V_{I_\theta}^\top w)$ and that $\nabla^2 L_\theta(w) = V_{I_\theta} \nabla^2 L(\theta + V_{I_\theta}^\top w) V_{I_\theta}^\top$. Thus we have that

$$(\nabla^2 L_\theta(0))^{-1} \nabla L_\theta(0) = V_{I_\theta} (\nabla^2 L(\theta))^{-1} \nabla L(\theta). \tag{45}$$

By the definition of $I_\theta$, we know that $\left\|V_{I_\theta} (\nabla^2 L(\theta))^{-1} \nabla L(\theta)\right\|_\infty \leq \rho$. Thus $\left\|(\nabla^2 L_\theta(0))^{-1} \nabla L_\theta(0)\right\|_2 \leq \sqrt{d} \left\|V_{I_\theta} (\nabla^2 L(\theta))^{-1} \nabla L(\theta)\right\|_\infty = \sqrt{d} \cdot \rho \leq \frac{R}{2}$. Thus we can apply Lemma G.8 on $L_\theta$ at $w = 0$ and conclude that

$$L_\theta(0) - L_\theta(w^*) \leq \nabla L_\theta(0)^\top (\nabla^2 L_\theta(0))^{-1} \nabla L_\theta(0) = \sum_{i \in I_\theta} \sigma_i^{-1} \left|v_i^\top \nabla L(\theta)\right|^2 \leq \Delta \tag{46}$$

Thus $L(\theta) - L(\theta + V_{I_\theta}^\top w^*) = L_\theta(0) - L_\theta(w^*) \leq \Delta$.

It remains to show that $L(\theta + V_{I_\theta}^\top w^*) - L(\theta^*) \leq \frac{25\Delta^2}{\rho^2 \mu}$. To do so, our strategy is to first show that $\left\|\nabla L(\theta + V_{I_\theta}^\top w^*)\right\|_2$ is small and then to use Lemma G.6. We will use $I_\theta^c$ to denote the complement of $I_\theta$ in $[d]$ and $V_{I_\theta^c} \in \mathbb{R}^{(d-|I_\theta|) \times d}$ to denote the submatrix of $V$ which contains all the rows that do not belong to $I_\theta$. Note that $w^*$ is the minimizer of $L_\theta$, we know that $V_{I_\theta} \nabla L(\theta + V_{I_\theta}^\top w^*) = 0$ and that $\left\|\nabla L(\theta + V_{I_\theta}^\top w^*)\right\|_2 = \left\|V_{I_\theta^c} \nabla L(\theta + V_{I_\theta}^\top w^*)\right\|_2$.

Now we consider the following ODE

$$\frac{\mathrm{d}w(t)}{\mathrm{d}t} = -(\nabla^2 L_\theta(w(t)))^{-1} \nabla L_\theta(0), \quad w(0) = 0. \tag{47}$$

By Lemma G.4, we know this ODE has solution $w(t)$ over interval $[0, 1]$ with $w(1) = w^*$. With the same argument in the proof of Lemma G.8, we know that $\|w(t)\|_2 \leq R$ for all $t \in [0, 1]$. Thus we

have for any $t \in [0, 1]$,

$$\left\| V_{I_\theta^c} \frac{\mathrm{d}\nabla L(\theta + V_{I_\theta} w(t))}{\mathrm{d}t} \right\|_2 \tag{48}$$

$$= \left\| V_{I_\theta^c} \nabla^2 L(\theta + V_{I_\theta} w(t)) V_{I_\theta} (\nabla^2 L_\theta(w(t)))^{-1} \nabla L_\theta(0) \right\|_2 \tag{49}$$

$$= \left\| V_{I_\theta^c} \nabla^2 L(\theta + V_{I_\theta} w(t)) V_{I_\theta} V_{I_\theta}^\top (\nabla^2 L(\theta + V_{I_\theta} w(t)))^{-1} \nabla L(\theta) \right\|_2 \tag{50}$$

$$\leq \left\| V_{I_\theta^c} \sqrt{\nabla^2 L(\theta + V_{I_\theta} w(t))} \right\|_F \tag{51}$$

$$\cdot \left\| \sqrt{\nabla^2 L(\theta + V_{I_\theta} w(t))} V_{I_\theta} V_{I_\theta}^\top (\nabla^2 L(\theta + V_{I_\theta} w(t)))^{-1} \nabla L(\theta) \right\|_2 \tag{52}$$

For the first term (Equation 51), by Assumption F.2, we have that

$$\left\| V_{I_\theta^c} \sqrt{\nabla^2 L(\theta + V_{I_\theta} w(t))} \right\|_F^2 \leq 2 V_{I_\theta^c} \nabla^2 L(\theta) V_{I_\theta^c} = 2 \sum_{i \notin I_\theta} \sigma_i \leq 2 \sum_{i \notin I_\theta} \frac{v_i^\top \nabla L(\theta)}{\rho} \leq \frac{2\Delta}{\rho^2}. \tag{53}$$

For the second term (Equation 52), by Assumption F.2, we have that

$$\left\| \sqrt{\nabla^2 L(\theta + V_{I_\theta} w(t))} V_{I_\theta} V_{I_\theta}^\top (\nabla^2 L(\theta + V_{I_\theta} w(t)))^{-1} \nabla L(\theta) \right\|_2^2 \tag{54}$$

$$\leq 8 \left\| \sqrt{\nabla^2 L(\theta)} V_{I_\theta} V_{I_\theta}^\top (\nabla^2 L(\theta))^{-1} \nabla L(\theta) \right\|_2^2 \tag{55}$$

$$= 8 \nabla L(\theta)^\top V_{I_\theta} V_{I_\theta}^\top (\nabla^2 L(\theta))^{-1} V_{I_\theta} V_{I_\theta}^\top \nabla L(\theta) \tag{56}$$

$$= 8 \sum_{i \in I_\theta} \sigma_i^{-1} \left| v_i^\top \nabla L(\theta) \right|^2 \leq 8\Delta. \tag{57}$$

Thus we conclude that $\left\| V_{I_\theta^c} \frac{\mathrm{d}\nabla L(\theta + V_{I_\theta} w(t))}{\mathrm{d}t} \right\|_2 \leq \frac{4\Delta}{\rho}$, which implies that

$$\left\| \nabla L(\theta + V_{I_\theta}^\top w^*) \right\|_2 = \left\| V_{I_\theta^c} \nabla L(\theta + V_{I_\theta}^\top w^*) \right\|_2 \tag{58}$$

$$= \left\| V_{I_\theta^c} \nabla L(\theta) + \int_{t=0}^1 V_{I_\theta^c} \frac{\mathrm{d}\nabla L(\theta + V_{I_\theta} w(t))}{\mathrm{d}t} \mathrm{d}t \right\|_2 \tag{59}$$

$$\leq \left\| V_{I_\theta^c} \nabla L(\theta) \right\|_2 + \int_{t=0}^1 \left\| V_{I_\theta^c} \frac{\mathrm{d}\nabla L(\theta + V_{I_\theta} w(t))}{\mathrm{d}t} \right\|_2 \mathrm{d}t \tag{60}$$

$$\leq \frac{\Delta}{\rho} + \frac{4\Delta}{\rho} = \frac{5\Delta}{\rho}. \tag{61}$$

Applying Lemma G.6, we have that

$$L(\theta + V_{I_\theta}^\top w^*) - \min L \leq \mu^{-1} \left\| \nabla L(\theta + V_{I_\theta}^\top w^*) \right\|_2^2 = \frac{25\Delta^2}{\rho^2 \mu}. \tag{62}$$

This completes the proof. $\qquad \square$

**Lemma G.10** (Descent Lemma). *For any $\eta, \rho > 0$ with $\eta\rho \leq R/\sqrt{d}$, $\theta \in \mathbb{R}^d$ and any eigendecomposition of $\nabla^2 L(\theta)$, where $V_t V_t^\top = I_d$, $\sigma_t$ is diagonal $\nabla^2 L(\theta) = V^\top \Sigma V$, define*

$$\theta_+ \triangleq \theta - \eta V^\top \mathrm{clip}(V(\nabla^2 L(\theta))^{-1}\nabla L(\theta), \rho), \tag{63}$$

*it holds that*

$$L(\theta_+) - L(\theta) \leq -(\eta - \eta^2) \sum_{i=1}^d \min\{\rho \left| v_i^\top \nabla L(\theta) \right|, \sigma_i^{-1} \left| v_i^\top \nabla L(\theta) \right|^2\}, \tag{64}$$

*where $v_i$ is the ith row of matrix $V$.*

*Proof of Lemma G.10.* Let $u \triangleq \mathrm{clip}(V(\nabla^2 L(\theta))^{-1}\nabla L(\theta), \rho)$. By the definition of clip operation, we know that $\left\| V^\top u \right\|_2 = \|u\|_2 \leq \sqrt{d}\rho$. Thus we have $\|\theta_+ - \theta\| = \eta \left\| V^\top u \right\|_2 \leq \eta\rho\sqrt{d}$. Define

$f(t) = L(t\theta_+ + (1-t)\theta)$. By Assumption F.2, we know that $f''(t) \leq 2f''(0)$ for all $t \in [0, 1]$ and thus

$$f(1) = f(0) + f'(0) + \int_{s=0}^{1} \int_{t=0}^{s} f''(s)\mathrm{d}s\mathrm{d}t \leq f(0) + f'(0) + f''(0). \tag{65}$$

It remains to show that

1.  $f'(0) = -\eta \sum_{i=1}^{d} \min\{\rho \left| v_i^\top \nabla L(\theta) \right|, \sigma_i^{-1} \left| v_i^\top \nabla L(\theta) \right|^2\}$;

2.  $f''(0) \leq \eta^2 \sum_{i=1}^{d} \min\{\rho \left| v_i^\top \nabla L(\theta) \right|, \sigma_i^{-1} \left| v_i^\top \nabla L(\theta) \right|^2\}$;

First, by chain rule, we have $f'(0) = \left\langle \nabla L(\theta), -\eta V^\top u \right\rangle = \left\langle V \nabla L(\theta), -\eta u \right\rangle = -\eta \left\langle V \nabla L(\theta), \mathrm{clip}(\Sigma^{-1} V \nabla L(\theta), \rho) \right\rangle = -\eta \sum_{i=1}^{d} \min\{\rho \left| v_i^\top \nabla L(\theta) \right|, \sigma_i^{-1} \left| v_i^\top \nabla L(\theta) \right|^2\}$. Second, again by chain rule, we have $f''(0) = \eta^2 \left\langle V^\top u, \nabla^2 L(\theta) V^\top u \right\rangle = \eta^2 \left\langle u, \Sigma u \right\rangle = \sum_{i=1}^{d} \left| u_i \right|^2 \sigma_i$. Note that by definition $\left| u_i \right| = \min\{\left| v_i^\top \nabla L(\theta) \right| / \sigma_i, \rho\}$, we have $\left| u_i \right|^2 \sigma_i \leq \min\{\left| v_i^\top \nabla L(\theta) \right| / \sigma_i, \rho\} \cdot \left| v_i^\top \nabla L(\theta) \right| / \sigma_i \cdot \sigma_i = \min\{\left| v_i^\top \nabla L(\theta) \right|^2 / \sigma_i, \rho \left| v_i^\top \nabla L(\theta) \right|\}$, which completes the proof. □

**Lemma G.11.** *If $\eta\rho \leq R/\sqrt{d}$ and for some $T \in \mathbb{N}$, $L(\theta_T) - \min L \leq \frac{\mu\rho^2}{8}$, then if holds that for all $t \geq T$,*

1.  $\theta_{t+1} = \theta_t - \eta(\nabla^2 L(\theta_t))^{-1} \nabla L(\theta_t)$;

2.  $L(\theta_t) - \min L \leq (1 - \eta(1-\eta))^{t-T}(L(\theta_T) - \min L)$.

*Proof of Lemma G.11.* First by Lemma G.10, we have for all $t \geq T$, $(\theta_t) - \min L \leq L(\theta_T) - \min L \leq \frac{\mu\rho^2}{8}$, therefore by Lemma G.7, we have $\left\| (\nabla^2 L(\theta_t))^{-1} \nabla L(\theta_t) \right\|_2 \leq \rho$ for all $t \geq T$, which implies clipping will not happen. This completes the proof of the first claim.

For the second claim, by Lemmas G.5 and G.10, we have that

$$L(\theta_{t+1}) - L(\theta_t) \leq -(\eta - \eta^2) \sum_{i=1}^{d} \sigma_i^{-1} \left| v_i^\top \nabla L(\theta_t) \right|^2 \tag{66}$$

$$= -(\eta - \eta^2) \nabla L(\theta_t)(\nabla^2 L(\theta_t))^{-1} \nabla L(\theta_t) \tag{67}$$

$$\leq -\eta(1-\eta)(L(\theta_t) - \min L), \tag{68}$$

which completes the proof. □

### G.1 LOWER BOUND FOR SIGNGD ON 2-DIMENSIONAL QUADRATIC LOSS

Define $L_{\mu,\beta} : \mathbb{R}^2 \to \mathbb{R}$ as a quadratic function with parameter $\mu, \beta$ as $L_{\mu,\beta}(\theta) \triangleq \frac{\mu}{2}\theta_{[1]}^2 + \frac{\beta}{2}\theta_{[2]}^2$. We have the following lower bound, which shows signGD's convergence rate has to depend on the condition number $\beta/\mu$.

**Theorem G.12.** *For any $\mu, \beta, \Delta, \epsilon > 0$, suppose there exist a learning rate $\eta$ and a time $T$ such that for all $\theta_0$ satisfying that $L_{\mu,\beta}(\theta_0) \leq \Delta$, signGD reaches loss at most $\epsilon$ at step $T-1$ and $T$ (in the sense that $L_{\mu,\beta}(\theta_T) \leq \epsilon$ and $L_{\mu,\beta}(\theta_{T-1}) \leq \epsilon$). Then, $T$ must satisfy $T \geq \frac{1}{2}(\sqrt{\frac{\Delta}{\epsilon}} - \sqrt{2})\sqrt{\frac{\beta}{\mu}}$.*

*Proof of Theorem G.12.* We consider two initialization: $\theta_0 = (0, \sqrt{\frac{2\Delta}{\beta}})$ and $\theta_0' = (\sqrt{\frac{2\Delta}{\mu}}, 0)$, and let $\theta_t$ and $\theta_t'$ be the iterates under the two initializations. For each coordinate $i \in \{1, 2\}$, because $\left| (\theta_t)_{[i]} - (\theta_{t+1})_{[i]} \right| = \eta$, we have that $\left| (\theta_t)_{[i]} \right| + \left| (\theta_{t+1})_{[i]} \right| \geq \eta$. Thus $2\epsilon \geq L_{\mu,\beta}(\theta_T) + L_{\mu,\beta}(\theta_{T-1}) \geq \frac{\beta}{2}((\theta_T)_{[2]}^2 + (\theta_{T-1})_{[2]}^2) \geq \frac{\beta\eta^2}{4}$, which implies $\eta \leq \sqrt{\frac{8\epsilon}{\beta}}$.

The fact that $L_{\mu,\beta}(\theta_T') + L_{\mu,\beta}(\theta_{T-1}') \leq 2\epsilon$ implies $(\theta_T')_{[1]} \leq \sqrt{\frac{4\epsilon}{\mu}}$. Because SignGD can only move each coordinate by $\eta$ at most, we have $(T-1)\eta \geq \sqrt{2\Delta/\mu} - \sqrt{\frac{4\epsilon}{\mu}}$. Using the fact that $\eta \leq \sqrt{\frac{8\epsilon}{\beta}}$, we have that $2(T-1) \geq (\sqrt{\frac{\Delta}{\epsilon}} - \sqrt{2})\sqrt{\frac{\beta}{\mu}}$, which completes the proof. □

## G.2 1D NON-CONVEX FUNCTION

In this subsection, we consider non-convex case. We start with an 1D non-convex function $L(\theta)$ where $\theta \in \mathbb{R}$.

For simplicity, we take $\epsilon = 0$ and consider the deterministic version of Sophia,

$$\theta \leftarrow \theta - \eta \cdot \text{clip}(\max\{L''(\theta), 0\}^{-1} \cdot L'(\theta), \rho). \tag{69}$$

where we use the convention that for any $x \in \mathbb{R}$, $x \neq 0$, $x/0 = \text{sign}(x) \cdot \infty$ and $0/0 = 0$.

For simplicity, we first assume that $L$ has a single minimizer (When there are more than one local minima, the algorithm will simplify converges to the closest local minima.)

**Assumption G.13.** *We assume that $L$ is a three times differentiable function with only one minimizer, $\theta^*$, where $\mu \triangleq L''(\theta^*) > 0$.*

The key challenges for second-order methods are two-fold as described below.

1. When the Hessian is much bigger than the gradient, then the update made by the second order algorithm $L''(\theta)^{-1}L'(\theta)$ is too small. For example, when $L(\theta) = \exp(\exp(\theta))$, second-oder algorithm cannot work. This oftentimes happens when the third-order derivatives is much larger than the second-order derivative, and as a consequence the second-order derivative is larger than the first-order derivative.

2. When there exists a saddle point where the gradient is zero, and but the Hessian is very close to zero. Such saddle points are often called non-strict saddle points Ge & Ma (2015).

The first situation is a pathological situation and past works on Newton's method oftentime assume that it does not occur. Situation 2 is also a common challenging situation that most prior works on non-convex optimization excludes Ge & Ma (2015); Lee et al. (2016); Jin et al. (2017) because SGD and many first order methods cannot escape such saddle points. Our assumptions below assume that these two cases do not occur.

**Definition G.14.** *For any $\gamma, G, \alpha \in \mathbb{R}^+$, we say that function $L : \mathbb{R} \to \mathbb{R}$ satisfies the $(\gamma, G, \alpha)$ strict-saddle-and-bounded-third-order-derivative condition iff*

$$G \leq \sup_{\epsilon > 0} \inf \left\{ |L'(\theta)| \ \Big| \ -\epsilon \leq L''(\theta) \leq \alpha \right\} \tag{70}$$

$$\gamma \geq \sup \left\{ 2\frac{|L'''(\theta)|}{L''(\theta)} \ \Big| \ L''(\theta) \geq \alpha/2 \right\} \tag{71}$$

Note that $\gamma > 0$ means that any stationary point has a decent Hessian, that is, is either a local minimum or local maximum with a reasonably strong curvature. This condition will matter more for the high-dimensional cases. Our bounds will depend on $G$ and $\gamma$ (though $G$ and $\gamma$ depends on $\alpha$ implicitly.) We also note that the bound (71) cannot hold over all $\theta \in \mathbb{R}$ (as opposed to over all $\theta$ such that $|L''(\theta)| \geq \alpha/2$) for non-convex function, because when $L''(\theta)$ changes sign, *i.e.*, from positive to negative, $\frac{|L'''(\theta)|}{L''(\theta)}$ has to be unbounded (or even undefined when $L''(\theta) = 0$.)

**Theorem G.15** (Main, 1D Non-convex). *Assume one-dimensional function $L$ satisfies Assumption G.13 and $(\gamma, G, \alpha)$ strict-saddle-and-bounded-third-order-derivative property (defined in Definition G.14). Then, picking $\eta = \frac{1}{2}$ and $\rho = \frac{1}{2\gamma}$, we have that the algorithm with in update Equation 69 reaches a loss at most $\epsilon$ in at most $O(T)$ steps where*

$$T = \frac{L(\theta_0) - \min L}{\min\{4\alpha/\gamma^2, G^2/\alpha\}} + \ln \frac{L(\theta_0) - \min L}{\epsilon} \tag{72}$$

## G.3 DECOMPOSABLE NON-CONVEX FUNCTIONS

Because our algorithm uses only element operations, it makes sense to consider a non-convex function $L(\theta)$ that can be decomposed into a sum of non-convex 1-D functions on each individual parameter:

$$L(\theta) = \sum_{i=1}^{d} L_i(\theta_{[i]}) \tag{73}$$

As have been done in Equation 69, for simplicity, we take $\epsilon = 0$ and consider the deterministic version of Sophia,

$$\forall i \in [d], \theta_{[i]} \leftarrow \theta_{[i]} - \eta \cdot \text{clip}(\max\{L_i''(\theta_{[i]}), 0\}^{-1} \cdot L_i'(\theta_{[i]}), \rho). \tag{74}$$

**Theorem G.16** (Non-convex, Decomposable). *Assume function $L$ is decomposable as in Equation 73. Suppose each $L_i$ satisfies Assumption G.13 with minimizer $\theta_{[i]}^*$ and $L_i''(\theta_{[i]}^*) = \mu_i$, has $(\gamma, G, \alpha)$ strict-saddle-and-bounded-third-order-derivative property (defined in Definition G.14). Then, picking $\eta = \frac{1}{2}$ and $\rho = \frac{1}{2\alpha}$, we have that the algorithm with in update Equation 74 reaches a loss at most $\epsilon$ in at most $O(T)$ steps where*

$$T = \frac{L(\theta_0) - \min L}{\min\{4\alpha/\gamma^2, G^2/\alpha\}} + \ln \frac{L(\theta_0) - \min L}{\epsilon} \tag{75}$$

We note that Theorem G.16 is a direct corollary of Theorem G.15. The Theorem does not depend on the condition number (the ratio between the largest and smallest curvatures among all dimensions). Instead, we assume that all dimension satisfy the same strict-saddle and bounded third-order derivative properties as stated in Definition G.14.

A strong theoretical analysis of any second-order optimizers' convergence for non-convex cases is of its own interest and we hope that our work can motivate more theoretical work on second-order algorithms in the future.

### G.4 PROOFS OF THEOREM G.15

We state a slightly more general version of Theorem G.15 for generality.

**Theorem G.17.** *Assume one-dimensional function $L$ satisfies Assumption G.13 and $(\gamma, G, \alpha)$ strict-saddle-and-bounded-third-order-derivative property (defined in Definition G.14). Then, for any $\eta < 1, \rho \leq \frac{1}{2\gamma}$, $\theta_0 \in \mathbb{R}$ and $\epsilon > 0$, it holds that*

$$\forall T \in \mathbb{N}, \quad T \geq \frac{1}{\eta(1-\eta)} \left( \frac{L(\theta_0) - \min L}{\min\{\alpha\rho^2, G^2/\alpha\}} + \ln \frac{L(\theta_0) - \min L}{\epsilon} \right) \implies L(\theta_T) \leq \epsilon. \tag{76}$$

*In particular, if we pick $\eta = \frac{1}{2}$ and $\rho = \frac{1}{2\gamma}$, we have that*

$$\forall T \in \mathbb{N}, \quad T \geq 4 \left( \frac{L(\theta_0) - \min L}{\min\{4\alpha/\gamma^2, G^2/\alpha\}} + \ln \frac{L(\theta_0) - \min L}{\epsilon} \right) \implies L(\theta_T) \leq \epsilon. \tag{77}$$

**Lemma G.18.** *Under Assumption G.13, for all $\theta \in \mathbb{R}$, $\langle \theta - \theta^*, L'(\theta) \rangle \geq 0$, and the equality is only attained at $\theta = \theta^*$.*

*Proof.* Lemma G.18 It suffices to show that there is only one stationary point, which is $\theta^*$. Suppose there is another stationary point which is larger than $\theta$, let $\theta'$ be the smallest stationary point larger than $\theta^*$. We have that $L'(\theta') - L'(\theta^*) = \int_{\theta=\theta^*}^{\theta'} L''(\theta) = 0$ and for any $\theta \in (\theta^*, \theta')$, $\int_{s=\theta^*}^{\theta} L''(s)\,ds$ has the same sign. The sign must be positive because $L''(\theta)$ is continuous and positive at $\theta^*$ by Assumption G.13. Thus we conclude that $L''(\theta') < 0$. However, this implies $\theta'$ is a local maximizer, which contradicts with Assumption G.13. □

**Lemma G.19.** *Under Assumption G.13, for any $\theta' \in \mathbb{R}$ such that $L''(\theta') < 0$, it holds that*

$$|L'(\theta')| \geq \sup_{\epsilon>0} \inf \left\{ |L'(\theta)| \, \Big| \, -\epsilon \leq L''(\theta) < 0 \right\}. \tag{78}$$

*Proof of Lemma G.19.* Since $L''$ is continuous, and $L''(\theta') < 0$, $L''(\theta^*) > 0$, we know for any $\epsilon > 0$, the set $\left\{ |L'(\theta)| \, \Big| \, -\epsilon \leq L''(\theta) < 0 \right\}$ is not empty. Now we claim that for any $\epsilon > 0$, there is a $\theta_\epsilon$ such that $|L'(\theta_\epsilon)| \leq |L'(\theta')|$ and $-\epsilon \leq L''(\theta_\epsilon) < 0$, which would imply that $|L'(\theta')| \geq \sup_{\epsilon>0} |L'(\theta_\epsilon)| \geq \sup_{\epsilon>0} \inf \left\{ |L'(\theta)| \, \Big| \, -\epsilon \leq L''(\theta) < 0 \right\}$.

Now we prove the above claim. Without loss of generality, we assume $\theta' > \theta^*$. By Lemma G.18, we know $L'(\theta)$ is positive on $(\theta^*, \infty)$. Now let $\Delta$ be the infimum of positives number such that

$L''(\theta^* + \Delta) = 0$. (if no positive $\Delta$ makes $L''(\theta^* + \Delta) = 0$, we set $\Delta = \infty$. ) It is evident that $L'(\theta)$ is positive and decreasing on $(\theta^*, \theta^* + \Delta)$. We proceed in two cases: (1) If $\Delta$ is not $\infty$, then since $L''$ is continuous, we know $L''(\theta^* + \Delta) = 0$ and $\Delta$ is the smallest number such that $L''(\theta^* + \Delta) = 0$. Thus for any $\epsilon > 0$, there is some point sufficiently close to $\theta^* + \Delta$ which satisfies the requirement in the claim. (2) If $\Delta = \infty$, note for any $\theta > \theta'$, we have that $-L(\theta') \leq L'(\theta) - L'(\theta') = \int_{s=\theta'}^{\theta} L''(s)\mathrm{d}s \leq (\theta - \theta') \cdot \sup_{\theta \in [\theta', \infty)} L''(\theta)$. Taking $\theta \to \infty$, we know that $\sup_{\theta \in [\theta', \infty)} L''(\theta) = 0$. Thus for any $\epsilon > 0$, there is some point larger than $\theta'$ but with arbitrarily small Hessian (in absolute value) which satisfies the claim. $\square$

As a direct corollary of Lemma G.19, we have Corollary G.20.

**Corollary G.20.** *Under Assumption G.13, for any $\gamma, G, \alpha \in \mathbb{R}^+$, and function $L : \mathbb{R} \to \mathbb{R}$ satisfies the $(\gamma, G, \alpha)$ strict-saddle-and-bounded-third-order-derivative condition*

$$G \leq \inf \left\{ |L'(\theta)| \ \Big| \ -\infty < L''(\theta) \leq \alpha \right\} \tag{79}$$

$$\gamma \geq \sup \left\{ 2\frac{|L'''(\theta)|}{L''(\theta)} \ \Big| \ L''(\theta) \geq \alpha/2 \right\} \tag{80}$$

**Lemma G.21.** *In the setting of Theorem G.17, for any $\theta, \theta' \in \mathbb{R}$ satisfying $|\theta - \theta'| \leq 1/\gamma$ ($1/\gamma = \infty$ if $\gamma = 0$), it holds that*

1. $L''(\theta) \geq \alpha \implies L''(\theta') \geq \frac{L''(\theta)}{2}$;

2. $L''(\theta) \geq \alpha/2 \implies L''(\theta') \leq 2L''(\theta)$;

*Proof of Lemma G.21.* We first define $f(t) \triangleq f(\theta + t\,\mathrm{sign}(\theta' - \theta))$. We will prove the first claim by contradiction. Suppose the first claim is not true, then there exists $t \in (0, 1/\gamma]$ such that $L''(f(t)) < \frac{L''(\theta)}{2}$. We let $t^*$ be the smallest number in $(0, 1/\gamma]$ that $L''(f(t^*)) = \frac{L''(\theta)}{2}$. Such $t^*$ always exists because $L''(\theta)$ is differentiable by Assumption G.13 and thus continuous. By definition of $t^*$, we know that for all $t \in [0, t^*]$, $\gamma/2 \geq \frac{|L'''(f(t))|}{L''(f(t))} \geq |\ln L''(f(t))|$. Thus $\ln 2 = \ln L''(f(t^*)) - \ln L''(f(0)) = \int_{t=0}^{t^*} \frac{\mathrm{d} \ln L''(f(t))}{\mathrm{d}t}\mathrm{d}t \geq t^*\gamma/2 \geq -1/2$. However, $-\ln 2 < -1/2$. Contradiction!

Suppose $L''(f(t)) \geq \alpha/2$ for all $t \in [0, 1]$, then we have $\ln L''(f(|\theta^* - \theta|)) - \ln L''(f(0)) = \int_{t=0}^{|\theta^* - \theta|} \frac{\mathrm{d} \ln L''(f(t))}{\mathrm{d}t}\mathrm{d}t \leq |\theta^* - \theta|\,\gamma/2 \leq 1/2 < \ln 2$. Otherwise, let $t^*$ be the largest number in $[0, 1)$ such that $L''(f(t^*)) = \alpha/2$. Applying the previous argument we have $L''(\theta') \leq 2L''(f(t^*)) = \alpha \leq L''(\theta)$. This completes the proof. $\square$

**Lemma G.22.** *In the setting of Theorem G.17, for any $\theta \in \mathbb{R}$ satisfying that $\left|((L''(\theta))^{-1}L'(\theta)\right|_2 \leq \frac{1}{2\gamma}$ and that $L''(\theta) > \alpha$, it holds that*

$$L(\theta) - \min L \leq (L''(\theta))^{-1} |L'(\theta)|^2 \leq 4(L(\theta) - \min L). \tag{81}$$

*Proof of Lemma G.22.* By Lemma G.21, we know that $L''(\theta') \geq \frac{1}{2}L''(\theta)$ for all $|\theta - \theta'| \leq 1/\gamma$. Thus we must have $\left\langle L'(\theta), L'(\theta - \mathrm{sign}(L'(\theta)) \cdot \frac{1}{\gamma}) \right\rangle \leq |L'(\theta)|^2 - |L'(\theta)|\frac{L''(\theta)}{2\gamma} < 0$. Thus $|\theta^* - \theta| \leq 1/\gamma$ and $L$ is $\frac{\nabla^2 L(\theta)}{2}$-strongly convex between $\theta$ and $\theta^*$, which implies that

$$L(\theta^*) \geq L(\theta) + (\theta^* - \theta)L'(\theta) + \frac{1}{4}|\theta^* - \theta|^2 L''(\theta) \geq L(\theta) - \frac{1}{4}|L'(\theta)|^2 (L''(\theta))^{-1}. \tag{82}$$

Similarly, by Lemma G.21, we know that $L''(\theta') \leq 2L''(\theta)$ for all $|\theta - \theta'| \leq 1/\gamma$. Thus we have

$$L(\theta^*) \leq L(\theta) + (\theta^* - \theta)L'(\theta) + |\theta^* - \theta|^2 L''(\theta) \leq L(\theta) - |L'(\theta)|^2 (L''(\theta))^{-1}. \tag{83}$$

The proof is completed by noting that $L(\theta^*) = \min L$. $\square$

**Lemma G.23** (Descent Lemma for 1D non-convex loss). *In the setting of Theorem G.17, for any $\eta, \rho > 0$ with $\eta\rho \leq 1/\gamma$ and $\theta \in \mathbb{R}$. Define*

$$\theta_+ \triangleq \theta - \eta\mathrm{clip}\left(\frac{L'(\theta)}{\max\{\nabla^2 L(\theta), 0\}}, \rho\right), \tag{84}$$

*it holds that*

$$L(\theta_+) - L(\theta) \leq -(\eta - \eta^2)\min\left\{\rho|L'(\theta)|, \frac{|L'(\theta)|^2}{\max\{L''(\theta), \alpha\}}\right\} \tag{85}$$

$$\leq -(\eta - \eta^2)\min\left\{\rho^2\alpha, G^2/\alpha, L(\theta) - \min L\right\} \tag{86}$$

*Proof of Lemma G.23.* We first prove (85). Let $u \triangleq \mathrm{clip}\left(\frac{L'(\theta)}{\max\{\nabla^2 L(\theta), 0\}}, \rho\right)$. By the definition of clip operation, we know that $|u| \leq \rho$. Thus we have $|\theta_+ - \theta| = \eta|u| \leq \eta\rho$. We will proceed by discussing two cases respectively: $L''(\theta) \geq \alpha$ and $L''(\theta) \leq \alpha$. Define $f(t) = L(t\theta_+ + (1-t)\theta)$.

First, for the case $L''(\theta) \geq \alpha$. By Lemma G.21, we know that $f''(t) \leq 2f''(0)$ for all $t \in [0,1]$ and thus

$$f(1) = f(0) + f'(0) + \int_{s=0}^1 \int_{t=0}^s f''(s)\mathrm{d}s\mathrm{d}t \leq f(0) + f'(0) + f''(0). \tag{87}$$

It remains to show that

1. $f'(0) = -\eta\min\{\rho|L'(\theta)|, (L''(\theta))^{-1}|L'(\theta)|^2\}$;

2. $f''(0) \leq -\eta^2\min\{\rho|L'(\theta)|, (L''(\theta))^{-1}|L'(\theta)|^2\}$;

The first claim is immediate by chain rule. The second claim holds because $uL''(\theta) \leq |L'(\theta)|$ by definition of $u$. This completes the proof of the first case.

For the second case, we have $L''(\theta) \leq \alpha$, which implies that $|u| \geq |u'|$ where $u' \triangleq \min\{\frac{|L'(\theta)|}{\alpha}, \rho\}$. We have $L(\theta_+) - L(\theta) = \int_{\theta'=\theta}^{\theta-u} L'(\theta')\mathrm{d}\theta' \leq \int_{\theta'=\theta}^{\theta-u'} L'(\theta')\mathrm{d}\theta'$, due to Lemma G.18. Furthermore, we have that

$$\int_{\theta'=\theta}^{\theta-\eta u'} L'(\theta')\mathrm{d}\theta' = -\eta u' L'(\theta) + \int_{t=\theta}^{\theta-\eta u'} \int_{s=\theta}^t L''(s)\mathrm{d}s\mathrm{d}t \leq -\eta u'L'(\theta) + \eta^2\alpha|u'|^2, \tag{88}$$

where the last we use $L''(s) \leq 2\alpha$ by Lemma G.21. The rest of the proof of the second case is the immediate and the same as that of the first case.

Now we turn to the proof of the simplified version (86). When $\rho|L'(\theta)| \leq \frac{|L'(\theta)|^2}{\max\{L''(\theta), \alpha\}}$, we have $|L'(\theta)| \geq \rho\alpha$. Thus $\min\left\{\rho|L'(\theta)|, \frac{|L'(\theta)|^2}{\max\{L''(\theta), \alpha\}}\right\} \geq \min\left\{\rho\alpha, \frac{|L'(\theta)|^2}{\max\{L''(\theta), \alpha\}}\right\}$. For the term $\frac{|L'(\theta)|^2}{\max\{L''(\theta), \alpha\}}$, if $L''(\theta) < \alpha$, then by Corollary G.20, we know $|L'(\theta)| \geq G$ and thus $\frac{|L'(\theta)|^2}{\max\{L''(\theta), \alpha\}} \geq \frac{G^2}{\alpha}$. Otherwise, if $L''(\theta) \geq \alpha$, then by Lemma G.22, we have $\frac{|L'(\theta)|^2}{\max\{L''(\theta), \alpha\}} \geq L(\theta) - \min L$. This completes the proof. $\square$

Now we are ready to prove Theorem G.17.

*Proof of Theorem G.17.* The theorem is a direct consequence of Lemma G.23 (Descent Lemma for 1D loss). $\square$

