# OpenReview forum: "Sophia: A Scalable Stochastic Second-order Optimizer for Language Model Pre-training"
_ICLR.cc/2024/Conference — ICLR 2024 poster_

### Official Review · Reviewer_Gbmx · 2023-10-31

**Soundness:** 4 excellent
**Presentation:** 2 fair
**Contribution:** 4 excellent
**Rating:** 8
**Confidence:** 4

**Summary:**

The paper proposes Sophia, a scalable optimizer with diagonal pre-conditioner and coordinate-wise clipping. Two estimators of the diagonal Hessian, Hutchinson and Gauss-Newton-Bartlett, are discussed. The authors illustrate the intuition behind Sophia via a simple yet convincing example. Some theoretical results are also derived. Numerical experiments are conducted on the pre-training of large language models to demonstrate the significant speed-up over other optimizers like AdamW and Lion.

**Strengths:**

Sophia proposes several enhancements on top of usual first-order optimizers. The use of diagonal Hessian exploits the curvature to accelerate convergence while keeping per-iteration cost comparable to first-order methods. Furthermore, the coordinate-wise clipping safeguards the iterates from unreliable second-order information and improves stability. The example and discussion in Section 2.1 illustrate the intuition behind Sophia in a very clear manner. The numerical experiments are comprehensive and the performance of Sophia looks quite strong.

**Weaknesses:**

- The writing and presentation have room for improvement. For example, the use of $L()$ and $l()$ are kind of messy. And function $\text{clip}()$ is used before its definition.

- There is a huge gap between Sophia and the theoretical results derived in the paper. The theoretical guarantee is for an algorithm which is (almost) completely different with Sophia under a fairly strong assumption.

- Further studies are required to show if Sophia's speedup still exists on real large language models with billions of parameters and models other than GPT. See Questions for more discussion on this point.

**Questions:**

Overall, I think this is an excellent work. Though the theoretical results are weak, the numerical performance of Sophia is significant and impressive. Here are several questions for clarification and further improvement.

- The authors mention that there is an efficient implementation in PyTorch and JAX for Hessian-vector product. What is the actual computational cost of this Hessian-vector product, compared with the computation of the (stochastic) gradient?

- Does the 2x speedup still exist when considering larger models typically with billions of parameters? Can authors add more explanation for Figure 10? I am also curious about the performance of Sophia on other models like LLaMA and BERT.

- This question might be a little beyond the scope of this work. Does Sophia still work well on other tasks beyond pre-training, such as fine-tuning? Can the authors provide some comment on this?

---

> ### Author Response · Authors · 2023-11-23
> **Response to Reviewer Gbmx**
>
> Thanks for the feedback! We address questions below:
>
> **Q1:** What is the actual computational cost of this Hessian-vector product, compared with the computation of the (stochastic) gradient?
>
>
> > **Response**: In practice, we observe that on a 1.5B model a Hessian-vector product with batchsize=64 costs approximately the same time as a stochastic gradient with batchsize=480.
>
>
>
>
> **Q2:** Does the 2x speedup still exist when considering larger models typically with billions of parameters?
>
>
> > **Response**: So far 1.5B and 6.6B GPT NeoX are the largest models which we have resources to run. We still observe 2x speedup of Sophia over Adam on the 1.5B model, but we still need more time and compute to complete the 6.6B run with more steps.
>
>
> **Q3:** Can authors add more explanation for Figure 10?
>
>
> > **Response**: We only carried two runs for AdamW and one run for Sophia on the 6.6B model. For AdamW, we tried peak learning rate 1.2e-4 (black solid curve) and 1.5e-4 (black dashed curve). The loss of the 1.5e-4 run blew up at about 17K steps. This indicates that 1.2e-4 is close to the largest peak learning rate for AdamW in that setup. Note that with the unstable 1.5e-4 learning rate, AdamW is slower than Sophia-G before blowing up.
>
>
>
>
>
>
> **Q4:** I am also curious about the performance of Sophia on other models like LLaMA and BERT.
>
>
> > **Response**: We provide results of mask language modeling with BERT in the table below. In terms of validation pre-training loss, Sophia is better than AdamW and Lion with less than 70% compute spent. We will provide results on T5 and contrastive pre-training in the next version.
>
>
> | Method  | Steps | Validation loss |
> | -------- | ------- | ------- |
> | Lion | 38400 | 1.620 |
> | AdamW | 38400 | 1.638 |
> | Sophia | 25600 | 1.605 |

---

### Official Review · Reviewer_knid · 2023-10-31

**Soundness:** 3 good
**Presentation:** 3 good
**Contribution:** 4 excellent
**Rating:** 8
**Confidence:** 3

**Summary:**

The authors proposed Sophia, a class of optimization methods making use of second-order information without dramatically increasing the computational cost. The paper focuses on two instantiations: Sophia-H and Sophia-G. In each algorithm, the second-order information comes from an estimation of the Hessian diagonal, which is updated by the moving average of the gradients divided by the moving average of the estimated Hessian, followed by a novel element-wise clipping procedure. In extensive numerical experiments, Sophia outperformed Adam by a 2x speed-up, making Sophia the first second-order optimizer achieving a speed-up on decoder-only large language models in wall-clock time or total compute.

**Strengths:**

The paper is well-organized and well-written. The contributions of this paper are significant. It is an interesting idea to use clipping to regularize the update size, in contrast to some prevalent choices like backtracking line-search. The method of updating the Hessian matrix, in which the matrix is updated every $k$ steps rather than every step, offers a tradeoff between Hessian estimation accuracy and computational cost. Moreover, Equation 10 provides a practical implementation of the GNB estimator. The paper also has extensive numerical experiments and some theoretical analysis, demonstrating the benefits of Sophia over other competitive methods. Furthermore, according to Figures 6 and 7, Sophia is insensitive to the choice of hyperparameters.

**Weaknesses:**

In the nonconvex setting, saddle points because very attractive to Newton-like optimizers, but it is non-obvious if Sophia can converge to a local minimum instead of a saddle point. While the authors provide an intuitive argument in Figure 2, it is not a rigorous one. The saddle-free Newton method [1] might be of some interest: it can additionally leverage the absolute curvature information when the Hessian diagonal is negative, in which case we also want to move fast.

[1] Identifying and attacking the saddle point problem in high-dimensional non-convex optimization (https://arxiv.org/abs/1406.2572)

**Questions:**

Adam is essentially RMSProp + momentum, where RMSProp is a method for adjusting the learning rate.
If my understanding is correct, Sophia is an entrywise learning rate scaling method, so a natural question is: have you considered combining Sophia with momentum?

How well does Sophia work for (potentially parameter-efficient) fine-tuning tasks instead of training a foundation model from scratch?

The idea of estimating the Hessian diagonal is not new, and both estimators proposed in the paper are not super novel. However, no second-order optimizer has performed as well as Sophia according to your experiments. How do you explain the success of Sophia? Is it due to regularization via gradient clipping?

---

> ### Author Response · Authors · 2023-11-23
> **Response to Reviewer knid**
>
> Thanks for noting that "The contributions of this paper are significant" and "It is an interesting idea to use clipping to regularize the update size".
>
> **Q1:** If my understanding is correct, Sophia is an entrywise learning rate scaling method, so a natural question is: have you considered combining Sophia with momentum?
>
>
> > **Response**: We'd like to clarify that Sophia uses exponential moving average (EMA) of stochastic gradients in the update rule as in the second paragraph of Section 2.2 in page 4 of the manuscript. We will make this more prominent in the revision.
>
>
> **Q2:** How do you explain the success of Sophia? Is it due to regularization via gradient clipping?
>
>
> > **Response**: As demonstrated in the ablation study (Section 3.5), element-wise update clipping and the Hessian estimators are both reasons for Sophia's success. Compared to AdaHessian, the Hutchinson estimator in Sophia-H has a better denoising effect. Compared with empirical Fisher, label resampling (equation 8) makes the GNB estimator of Sophia-G an unbiased estimator of the Gauss-Newton term. Element-wise clipping allows infrequent updates of the diagonal Hessian, which reduces the overhead of computing second-order derivatives.

---

### Official Review · Reviewer_x5VB · 2023-11-03

**Soundness:** 3 good
**Presentation:** 3 good
**Contribution:** 2 fair
**Rating:** 6
**Confidence:** 4

**Summary:**

The paper describes a second order optimizer Sophia. Sophia works by estimating the diagonal of the Hessian, and using it to precondition the gradient, in the family of Newton optimizers. The two main differences with previous similar work (Adahessian) are that Sophia applies per-parameter clipping to the Hessian to limit the effect of Hessian noise, and uses an exponential moving average (like Adam) to effectively compute diagonal Hessians over a larger batchsize, to reduce noise in the approximation.

The authors mention two methods to estimate the diagonal Hessian --- the standard Hutchinson method that uses Hessian-vector products, and a relatively new Gauss-Newton-Bartlett method that uses a second forward-backward pass. The latter produces better results in their experiments. To mitigate the extra time needed for the diagonal Hessian computations, they use the standard heuristic of computing the diagonal Hessian infrequently, they compute it once every 10 steps.

The authors test their method by training GPT2 models with up to 6.6B parameters, showing that Sophia is able to achieve significant savings over Adam, which is commonly used to train GPT models. They also show that the model trained with Sophia for the same number of steps gets better accuracy on downstream SuperGLUE tasks than the model trained by AdamW for the same number of steps.

**Strengths:**

The novel contribution of this paper is to combine diagonal Hessian estimates (used by others, notably Adahessian), with exponential moving average of the Hessian and per parameter clipping of the Hessian (both also used before). The clipping allows them to reduce the effect of Hessian noise, and allows them to compute the Hessian infrequently, so the overhead of Hessian computation is not very large. The EMA improves stability of the Hessian computation.

The experiment with various versions of GPT seems quite clear, in all cases Sophia reached the same validation loss as Adam in fewer steps.

**Weaknesses:**

The main weakness is that the experiments are only in one problem domain --- decoder only LLMs of various sizes, although this is indeed a big application. Other problems and models should also be considered to see if the optimizer produces good results on different tasks.

The section 2.3 on GNB was not so clear, perhaps a more complete explanation in the appendix would help.

There is only weak theoretical justification for the optimizer, so it is justifiably relegated to the appendix.

**Questions:**

You said in the text that "the gap between Sophia and Adam with 100K steps increases as the model size increases", but the results in Figure 4 do not seem to support this claim --- the loss reached by Adam in 200K steps was reached by Sophia in about 160K steps in the 1.5B model, and similarly for the 6.6B model.

---

> ### Author Response · Authors · 2023-11-23
> **Response to Reviewer x5VB**
>
> Thanks for the feedback!
>
> **Q1: The experiments are only in one problem domain --- decoder only LLMs of various sizes, although this is indeed a big application. Other problems and models should also be considered to see if the optimizer produces good results on different tasks.**
>
>
> > **Response**:
> We provide results of mask language modeling with BERT in the table below. In terms of validation pre-training loss, Sophia is better than AdamW and Lion with less than 70% compute spent.
>
>
> | Method  | Steps | Validation loss |
> | -------- | ------- | ------- |
> | Lion | 38400 | 1.620 |
> | AdamW | 38400 | 1.638 |
> | Sophia | 25600 | 1.605 |
>
>
> **Q2:** "the gap between Sophia and Adam with 100K steps increases as the model size increases", but the results in Figure 4 do not seem to support this claim --- the loss reached by Adam in 200K steps was reached by Sophia in about 160K steps in the 1.5B model, and similarly for the 6.6B model.
>
>
> > **Response**: Thanks for raising this point! Note that experiments of 125M, 355M, and 770M in Figure 4 (a)(b)(c) are carried out on OpenWebText with GPT-2, while those of 1.5B and 6.6B are on the Pile with GPT NeoX. The difference in datasets and architecture means that we cannot build a scale law with all the 5 scales. ,as a result, don't have 1.5B and 6.6B data points in the scaling law figure (Figure 1 (d)). We will modify the sentence to "On OpenWebText, the gap between Sophia and Adam with 100K targeted steps is larger on 355M / 770M models than 125M models" to make it more precise in the revision.
>
>
> **Q3:** The section 2.3 on GNB was not so clear, perhaps a more complete explanation in the appendix would help.
>
>
> > **Response**: We added an expanded version in the appendix.

---

### Official Review · Reviewer_R1G3 · 2023-11-06

**Soundness:** 3 good
**Presentation:** 4 excellent
**Contribution:** 4 excellent
**Rating:** 8
**Confidence:** 3

**Summary:**

The paper introduces a new optimization algorithm for pre-training LLMs. Their algorithm combines several standard techniques in an interesting way to come up with a novel update rule. Specifically, the techniques combined include: Hessian pre-conditioning of the gradient, coordinate-wise clipping of values in the update step, the use of the Gauss-Newton-Bartlett estimator for the Hessian's diagonal entries (justified by previous works the authors cite), and the idea of performing approximate computations each time and more correct computations only periodically. While each of these techniques is itself quite classical, the paper's combination of these ideas to give a new algorithm that does quite well experimentally is very impressive.

**Strengths:**

I believe the novelty and creativity of the paper's algorithm for what is currently a very relevant topic of study is its biggest strength. What I mean by novelty/creativity is the update step of  $$\max( \min ( \frac{\hat{g}}{\max( \hat{H} , \epsilon)} , 1 ), -1  ),$$ where $\hat{H}$ and $\hat{g}$ are the approximate Hessian and gradient, respectively. This formulation in some sense interpolates between using a Newton step and using SignSGD based on the local geometry of the loss function, thereby choosing the "right technique" between these two and avoiding the other's disadvantages (for example, avoiding getting too slow under heterogeneous curvatures due to Adam and avoiding getting stuck at a local-non-minimum due to Newton). In order to achieve this goal, the authors come up with computationally light estimators of the diagonal entries of the Hessian.

Through their above innovation, the authors make *second-order* methods computationally tractable for pre-training LLMs, which is a big achievement.

**Weaknesses:**

Please see Q1 and Q2 below. In my view, the paper could be stronger were these questions to be satisfactorily addressed.

**Questions:**

Thank you for an overall interesting and nicely written paper. I'd be interested in understanding the following.

**Question 1.** The theoretical analysis (Section E, F) of the submission seem to make assumptions of convexity and slow-changing Hessian on the loss function. This, to me, seems somewhat at odds with the motivation for the key novelty of the algorithm, which is to clip the value of the positively clipped Hessian (from my understanding, based on Page 3, "Limitations of Newton's method" paragraph, it looks like the role of the clipped Hessian is precisely to "mitigate the rapid change of Hessian" and be less "vulnerable to negative curvature" (cf. Section 2.2)). From this, it seems to me that the theory in Sections E and F are perhaps justifying only the gradient clipping, not the clipping under nonconvexity or negative-curvature-Hessian. Do the authors have any additional theory for the algorithm when the loss is nonconvex and/or when the Hessian has a negative curvature or is rapidly changing? This would, in my opinion, truly complement what's in the experiments and greatly strengthen the paper.

**Question 2.** The authors mention that their algorithm avoids local maxima and saddle points. My question is, would they be able to comment on what kind of local minima their algorithm converges to? Specifically, are these flat minima or sharp minima? Given that there has been a fair amount of work on the varying generalizability of these two types of minima, perhaps it might be important to understand this question so as to avoid overfitting.

**Comment 1.** I found Figure 1 too small to see the details. Since the authors use this figure to explain much of the initial intuition, it would help to have it bigger. (I understand space is limited, so I'm definitely not penalizing this aspect; but perhaps there could be a bigger version of the same figure in the appendix, or perhaps the authors could use another visual medium to show the phenomenon of Fig. 1.)

**Remark 1.** I would be happy to increase my score of "Soundness" and my "Confidence" were Q1 and Q2 to be satisfactorily answered. Regardless, I think this is a very good contribution to the ICLR community.

**Details Of Ethics Concerns:**

The paper is concerned with LLMs, particularly their fast pre-training. I don't have any specific concerns, but given that LLMs constitute an emerging technology with several nascent ethical issues, I recommend this paper for an ethics review.

---

> ### Author Response · Authors · 2023-11-23
> **Response to Reviewer R1G3**
>
> Thanks for noting the "novelty and creativity" of the paper and that "the authors make second-order methods computationally tractable for pre-training LLMs, which is a big achievement". We answer the questions as follows:
>
> **Q1:** The theoretical analysis seems to make assumptions of convexity and slow-changing Hessian on the loss function. This, to me, seems somewhat at odds with the motivation for the key novelty of the algorithm, which is to clip the value of the positively clipped Hessian. Do the authors have any additional theory for the algorithm when the loss is nonconvex and/or when the Hessian has a negative curvature or is rapidly changing?
>
>
> > **Response**: We agree that *Assumption E.2* requires the curvature to change at a reasonable rate. However, this assumption contains natural cases which do not satisfy the self-concordance assumptions [1] or bounded third-order derivatives assumptions [2] in Newton's method analysis. For example, $f(x) = \log(1+e^x)-x/2$ is not self-concordant (when x goes to infinity) but satisfies *Assumption E.2*. When x is very large, the hessian of $f$ is very small, but the gradient is close to 1. Newton's method will overshoot, but with clipping Sophia is fine with this case. For decomposable nonconvex functions, we provide analysis in Appendix G of the revision. We leave analysis for more general non-convex functions to future work.
>
>
> [1] Boyd, S. P. and Vandenberghe, L. Convex optimization.
> [2] Nesterov, Y. and Polyak, B. T. Cubic regularization of newton method and its global performance
>
>
> **Q2:** My question is, would they be able to comment on what kind of local minima their algorithm converges to? Specifically, are these flat minima or sharp minima?
>
>
> > **Response**: We did not observe a significant trend of the trace of Hessian of the pre-training loss of models trained with Sophia-G and AdamW and Lion as shown in the table below on 125M and 355M models pre-trained for 100K steps. In language model pre-training, we typically have strict data deduplication and pass one data point only once. Therefore the training loss and validation loss are almost the same and we are not overfitting training data. Performance on few-shot evaluation in Figure 5 also indicates the models trained with Sophia are transferable to downstream tasks.
>
>
> | Method  | Size | Trace of Hessian |
> | -------- | ------- | ------- |
> | Lion | 125M | 23.40 |
> | AdamW | 125M | 24.16 |
> | Sophia | 125M | 20.72 |
> | Lion | 125M | 29.95 |
> | AdamW | 125M | 27.37 |
> | Sophia | 125M | 28.16 |

---

### Meta-Review · Area_Chair_BeFN · 2023-12-03

**Metareview:**

The paper presents Sophia, a scalable optimizer with a diagonal pre-conditioner and coordinate-wise clipping. The estimate of the diagonal Hessian is used as preconditioner for which two estimators, Hutchinson and Gauss-Newton-Bartlett, are discussed. Although the theoretical guarantees are very limited and rely on rather restrictive assumptions, numerical experiments are convincing and demonstrate speed-ups over other optimizers. Overall, the paper addresses a timely problem and contains several interesting ideas and results.

**Justification For Why Not Higher Score:**

The theoretical guarantees are very limited in their scope and rely on rather restrictive assumptions.

**Justification For Why Not Lower Score:**

The paper is timely, well written, contains interesting ideas and numerical experiments are convincing and demonstrate speed-ups over other optimizers.

---

### Decision · Program_Chairs · 2024-01-16

Accept (poster)